# Coastal sedimentation across North America doubled in the 20th century despite river dams

A. B. Rodriguez [1,2 ✉], B. A. McKee[2], C. B. Miller [1,2], M. C. Bost[1,2] & A. N. Atencio[2]

The proliferation of dams since 1950 promoted sediment deposition in reservoirs, which is thought to be starving the coast of sediment and decreasing the resilience of communities to storms and sea-level rise. Diminished river loads measured upstream from the coast, however, should not be assumed to propagate seaward. Here, we show that century-long records of sediment mass accumulation rates (g cm$^{-2}$ yr$^{-1}$) and sediment accumulation rates (cm yr$^{-1}$) more than doubled after 1950 in coastal depocenters around North America. Sediment sources downstream of dams compensate for the river-sediment lost to impoundments. Sediment is accumulating in coastal depocenters at a rate that matches or exceeds relative sea-level rise, apart from rapidly subsiding Texas and Louisiana where water depths are increasing and intertidal areas are disappearing. Assuming no feedbacks, accelerating global sea-level rise will eventually surpass current sediment accumulation rates, underscoring the need for including coastal-sediment management in habitat-restoration projects.

[1] Institute of Marine Sciences, University of North Carolina at Chapel Hill, 3431 Arendell St., Morehead City, NC 28557, USA. [2] Department of Marine Sciences, University of North Carolina at Chapel Hill, Chapel Hill, NC 27599-3300, USA. ✉email: abrodrig@email.unc.edu

Sediment supply to nearshore areas can play a major role in determining the future of coastal communities and eco-systems threatened by accelerating sea-level rise (SLR) and increasing storminess. Intertidal habitats such as oyster reef, salt marsh, and mangrove provide coastal communities some protection from storms, are essential for the maintenance of healthy estuaries, and require allogenic sediment to sustain areal extent with SLR[1]. Along coastlines where sea level is rising, intertidal habitats could transition to subtidal if sedimentation cannot keep pace with SLR[2,3]. Large areas of intertidal oyster reef and salt-marsh have already been lost in North America[4,5]. Those losses, compounded with the coastal zone experiencing higher rates of population growth, urbanization, and investment compared to inland areas[6], exacerbates coastal communities' exposure to risk from natural hazards. Habitat restoration is commonly used to offset environmental degradation; however, with accelerating SLR[7] the initial success of restoration projects could decline if not supplemented with sediments after construction. The research community has emphasized regional monitoring, modeling, and forecasting of changes in sea level to improve coastal risk assessments but seldom (except for many wetland studies[8,9]) couple that information with regional measurements of sedimentation[10].

In North America, initial land-use change in upstream watersheds generally has caused river-sediment loads to increase from accelerated soil erosion, such as converting forest to farms[11]. Subsequently, watershed management has reduced surface runoff, increased the number of dams constructed, and disconnected upstream sediment sources and river loads from the coast[11–14]. Extensive river and watershed modifications can occur within decades. For example, the number of dams constructed in the United States increased from 16,000 by 1949 to 48,000 by 1969, mostly upstream of the low-gradient coastal plain[15]. Impoundments along large river systems, those that discharge directly into oceans and form deltas (e.g. Nile, Niger, and Mississippi rivers), are highlighted as major drivers for the post 1950 global reduction in river supply to the oceans[14,16]. Reduced sediment supply to modified large rivers has prevented some associated deltas from keeping pace with relative SLR, resulting in wetland loss through submergence and shoreline erosion[16]. Likewise, impoundments along smaller river systems in North America, those that discharge directly into estuaries and form bayhead deltas, drastically reduced suspended sediment loads measured at the furthest downstream gauge after 1950[12]. Gauges positioned furthest downstream are still far from the coast, located typically ~15–100 km landward of bayhead deltas. It is assumed, however, that the reduction in river load observed at downstream gauges is propagated further downstream, corresponds with a reduction in suspended sediment delivery to coastal areas, and will hasten the degradation of intertidal habitats with accelerated relative SLR[12,17].

The objective of this study is to test the assumption that North American coastal depocenters are sediment starved from the damming of rivers[12,17]. For this study, we define coastal depocenters as subtidal basins away from shorelines that are net depositional. We hypothesize that reduced suspended-sediment delivery to the coast from impoundments will be recorded in coastal depocenters as decreasing sedimentation rates over the last 100 years. Alternatively, if sediment sources positioned downstream of dams offset sediment lost to reservoirs[18], then that will be recorded as constant or increasing sedimentation rates. While dams certainly interrupt the river-sediment transport pathway, our results show that sedimentation in coastal depocenters more than doubled after 1950, likely due to supplemental sediment from human-modified landscapes. Sedimentation is keeping pace with relative SLR in many of the coastal depocenters, but not along rapidly-subsiding nearshore areas that are losing intertidal habitats, likely exemplifying the future broader region with accelerating SLR.

## Results and discussion

**Sedimentary record of coastal depocenters.** To assess coastal sedimentation around North America, we developed records of mass accumulation rates (MAR; g cm$^{-2}$ yr$^{-1}$) and sediment accumulation rates (SAR; cm yr$^{-1}$) from 25 coastal sites using published and new sediment-core geochronologies (Fig. 1; see "Methods" section and Supplementary Note 1 and Supplementary Data 1). The compilation of sites spans a wide range of geologic and climatic settings and includes estuaries (18; average core water depth = 10 m) and inner continental shelves (7; average core water depth = 49 m; Fig. 1; Supplementary Data 1). We used depocenters as archives of regional changes in coastal sedimentation, targeting sites where variations in those depositional, erosional, and mixing processes that form sedimentary records were generally constant through time (Supplementary Note 1). This avoids the tight coupling between intertidal sedimentation and inundation time[19], shoreline areas and frequent reworking[20], and river outlets and rapid regression or transgression of depositional environments[21].

The bathymetry of coastal depocenters is dynamic, deepening, and shallowing with changes in sedimentation, erosion, and relative sea level. The sedimentary record forms as sediment accommodation increases, which is the amount of space that is available for sediments to accumulate[22]. Sediment accommodation in coastal depocenters changes over long-time scales (years to millennia) by relative sea level, the morphology of coastal basins, and sedimentation[23]. Sediment supply is primarily driven over short time scales (days to years) by storm precipitation (flooding events), and over long-time scales by tectonics, climate change, and human modifications to rivers and watersheds. Any sediment deposition in excess of accommodation is above the depth of erosion, temporary, and resuspended during storms[24,25]. Relative SLR promotes sediment accumulation by shifting wave base above the bed, increasing accommodation. Over long-time scales sediment accumulation in coastal depocenters commonly matches relative SLR[26] and bathymetry remains constant; however, sediment supply has been severely impacted by humans.

The depocenters in our study are fed by rivers that contain dams[15] and primarily receive suspended sediment from numerous watersheds, including both small (<250 km$^2$) watersheds that are isolated to coastal regions and larger watersheds that extend into piedmont or high-relief areas. The sites on the middle continental shelf (sites 23 and 25) are more marine influenced than the other sites and receive some sediment from the along-shelf sediment dispersal system, making their sediments an indirect recorder of changes to watersheds (Fig. 1). A sediment budget for each depocenter that quantifies the relative contribution of sediment sources to the sedimentary record neither exists for the sites nor can be constructed from the existing data sets. The watersheds that contribute suspended sediment to a depocenter are modified by humans to different degrees with some of the smaller watersheds having few or no dams and the larger watersheds being severely modified by impoundments. Dam construction peaked after 1950 and reduced fluvial suspended sediment concentrations in the rivers away from the coast[12]. The connectivity of watersheds that contain dams to a depocenter varies among the sites. For example, the Nastapoka Sound depocenter is positioned 350 km down drift from the nearest outlet of a dammed river, in contrast to the 13 depocenters located in drowned river mouth estuaries that are more confined repositories for suspended sediment from

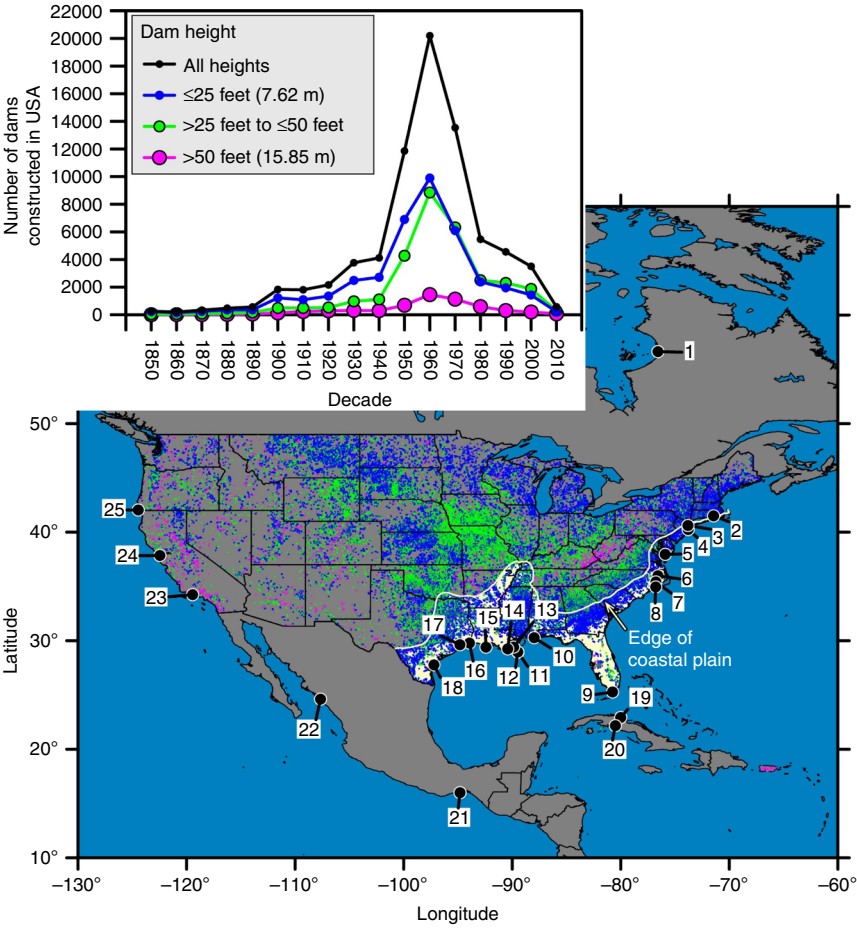

**Fig. 1 Location of study sites in North America and dams in the USA.** Most larger dams (green and magenta) are located inland from the coastal plain (yellow) and were constructed after 1950. Dam information from the Army Corps of Engineers National Inventory of Dams[15]. (1) Nastapoka Sound, Hudson Bay, Canada, (2) Pettaquamscutt River Basin, RI, USA, (3) Jamaica Bay, NY, USA, (4) New York Bight, NY, USA, (5) Pocomoke Sound, Chesapeake Bay, VA, USA, (6) Albemarle Sound, NC, USA, (7) Pamlico River Estuary, NC, USA, (8) Neuse River Estuary, NC, USA, (9) Florida Bay, FL, USA, (10) Mobile Bay, AL, USA, (11) Inner-Continental Shelf B, LA, USA, (12) Inner-Continental Shelf E, LA, USA, (13) Barataria Bay, LA, USA, (14) Terrebonne Bay, LA, USA, (15) Atchafalaya Delta, LA, USA, (16) Sabine Lake, TX, USA, (17) Galveston Bay, TX, USA, (18) Corpus Christi Bay, TX, USA, (19) Sagua Estuary, Cuba, (20) Cienfuegos Bay, Cuba, (21) Gulf of Tehuantepec, Mexico, (22) Culiacan River Estuary, Mexico, (23) Santa Clara Shelf, CA, USA, (24) San Francisco Bay, CA, USA, (25) Pacific Northwest Margin, OR, USA. See Supplementary Data 1 and Supplementary Note 1 for details on each site.

dammed rivers. Sedimentation at the sites before and after 1950 are compared because in North America this date generally represents an important shift where humans begin to make an indelible mark on landscapes and processes, as populations expanded[27]. In addition to the escalation in dam construction around 1950, the population of bordering coastal counties at all sites persistently increased since 1900 and doubled from 1950 to 2010 at 16 of the sites (Fig. 2; see "Methods" section).

Sedimentation is measured over the last 150 years by establishing geochronologies using the radioisotope $^{210}$Pb (half-life of 22.3 years). For watersheds, this isotope is primarily produced in the atmosphere, delivered to the surface via dust and precipitation, rapidly becomes irreversibly fixed to clay and silt, and follows the pathway of the suspended-sediment particles. A possible complication for establishing $^{210}$Pb geochronologies may arise at sites along the Pacific margin (sites 23 and 25) where the source of $^{210}$Pb from upwelling waters may vary (see Methods). As the particles accumulate over time in coastal sedimentary basins, the concentration of $^{210}$Pb decreases at a known decay rate to background levels at some depth that is determined, in part, by the sedimentation rate. Many of the published geochronologies that we utilize in this study were originally derived to calculate an average flux rate of particulates including

contaminants, organic carbon, and microfauna to coastal depocenters (see "Methods" section and Supplementary Note 1 for details on the sedimentology and geochemistry of the cores). Using those published $^{210}$Pb data to examine varying rates of sedimentation requires an alternative long-established modeling approach[28–30]. That modeling approach stipulates that the down-core distribution of $^{210}$Pb adheres to a well-defined set of criteria. These criteria, in part, limited the number of sites we could include in this survey (see "Methods" section).

**Mass accumulation rates.** The variety of settings included in the survey provides the dataset diversity in climate, sediment type, source area, depositional processes, and erosional processes, which drives the observed two-orders-of-magnitude range in MAR values (Figs. 3, 4) and give credence to the patterns and trends extracted from the dataset. For example, the main sediment source for sub-polar Nastapoka Sound in Hudson Bay is permafrost decay in adjacent river catchments[31], for arid Corpus Christi Bay, TX it is fluvial sediment from the small (39,000 km$^2$) Nueces River drainage basin and erosion of underlying Pleistocene strata[32], and for tropical Florida Bay, it is in-situ carbonate production[33]. Despite the large range of MAR values, most sites show a general increase through time (Figs. 3, 4). Sites with rapid sedimentation and

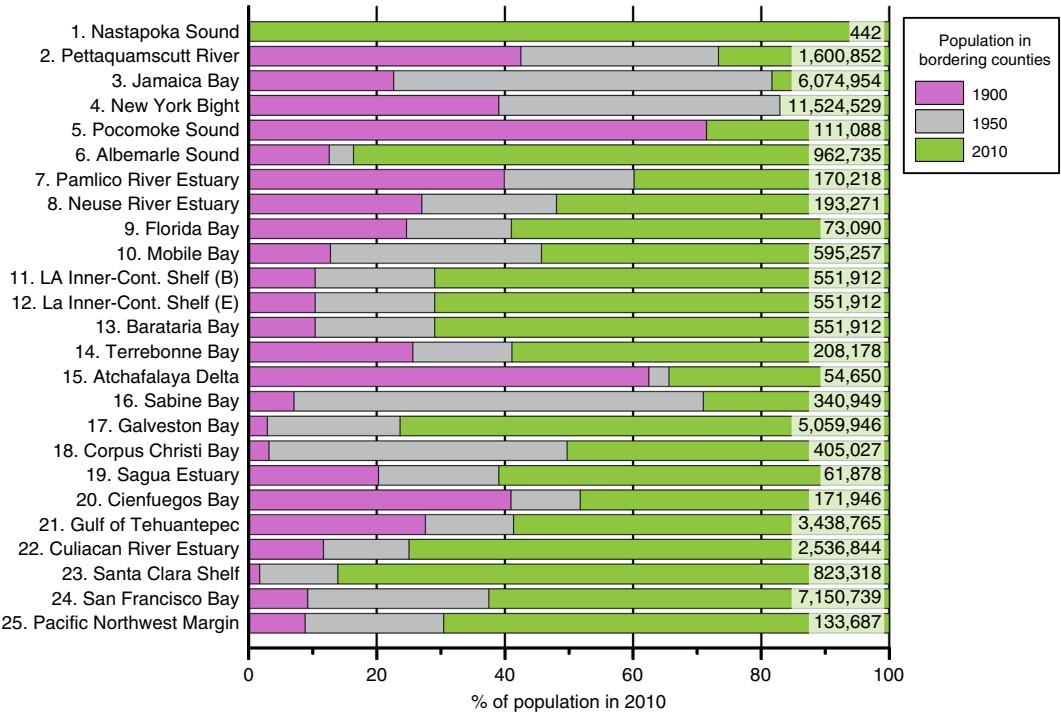

**Fig. 2 Coastal population growth since 1900.** The coastal population of counties adjacent to the sites increased sharply around 1950. Population data adjacent to Nastapoka Sound is only available for 2010. Population of counties adjacent to Pocomoke Sound decreased 2% in 1950. All populations in 2010 are enumerated. Data and sources provided in Supplementary Data 1.

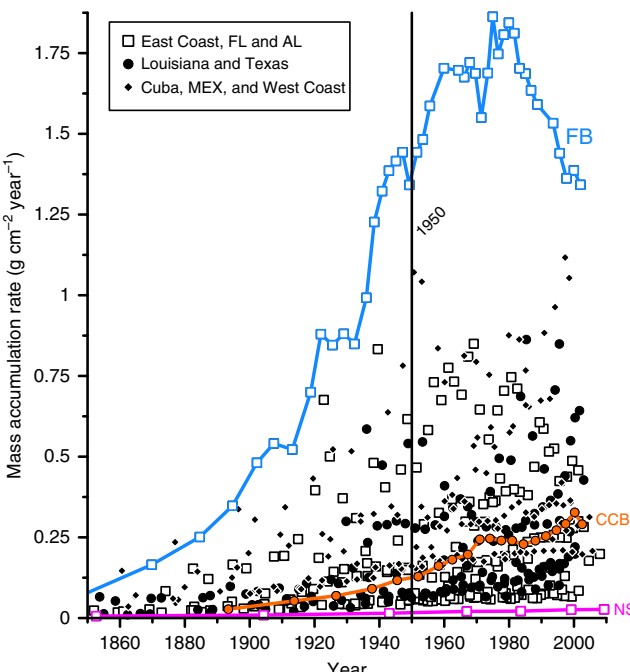

**Fig. 3 Mass accumulation rates among sites and through time.**
Sedimentary records ($N = 25$) extend >100 years, span a range of mass accumulation rates (MAR), and show an increase in sedimentation through time, as exemplified by Florida Bay (FB; average MAR = 1.405 g cm$^{-2}$ yr$^{-1}$), Corpus Christi Bay (CCB; average MAR = 0.195 g cm$^{-2}$ yr$^{-1}$), and Nastapoka Sound (NS; average MAR = 0.018 g cm$^{-2}$ yr$^{-1}$). Data sources can be found in Supplementary Note 1, and processed results in Supplementary Data 2.

associated higher resolution exhibited some decadal variability, the most extreme being Florida Bay where sediment MAR increased disproportionately from 1950 to 1980 with a subsequent decrease due to changes in Everglades water management that increased freshwater delivery and reduced macroalgae productivity, the main source of carbonate sediment to the depocenter[33,34].

The sediment geochronologies show post 1950 median MAR and first-quartile values increased above the pre 1950 median MAR and third-quartile values at most of the sites. The exceptions being the shelf near the Atchafalaya Delta where the river network is highly modified[35] and Sabine Bay where the margins of the depocenter are becoming subtidal[36] (Fig. 4). Similarly, average sediment MAR since 1950 increased at most of the sites as compared with average pre1950 values (median $p$-value = $3.92 \times 10^{-5}$), the only exception being Sabine Bay where the increase was not significant ($p$-value = 0.15; Fig. 5a; Supplementary Data 1). The land between Sabine and Calcasieu bays is being converted to open water, 24.5% submerged during the period 1956–2010[36]. For the MAR in Sabine Bay to remain constant over this period of depocenter expansion requires an increase in sediment flux. Like Sabine Bay, 25.9% and 24.5% of land was converted to open water after 1950 in Barataria and Terrebonne basins, respectively[36]. At those sites, average MAR more than doubled post 1950 because increased sediment supply to the depocenters exceeded the increase in subtidal areas.

**Downstream sediment sources to coastal depocenters.** The researchers who collected and processed the cores did not report an increase in the depth of surface mixing, grain size, bioturbation, and/or the density of accumulating particles, factors that could explain the magnitude of the increase in MAR (see Supplementary Note 1). This implies that the MAR increase, observed on continental shelves and estuaries, was caused by an increase in

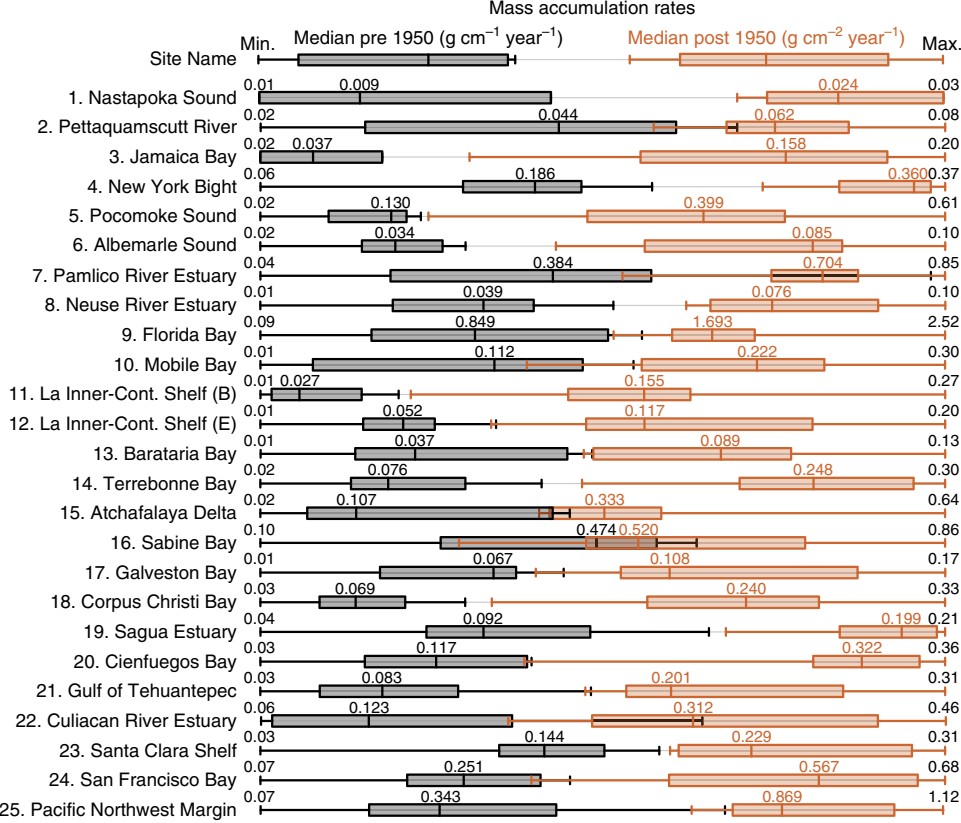

**Fig. 4 Summaries of pre and post 1950 mass accumulation rate bins.** The post 1950 median mass accumulation rate (MAR; brown) is higher than the pre 1950 median MAR at all sites (black; enumerated). The boxes represent the first and third quartiles and the whiskers show the minimum and maximum MAR for the pre and post 1950 bins. Upper and lower extreme MAR values for the entire time series are enumerated for each site. See Supplementary Data 2 for details.

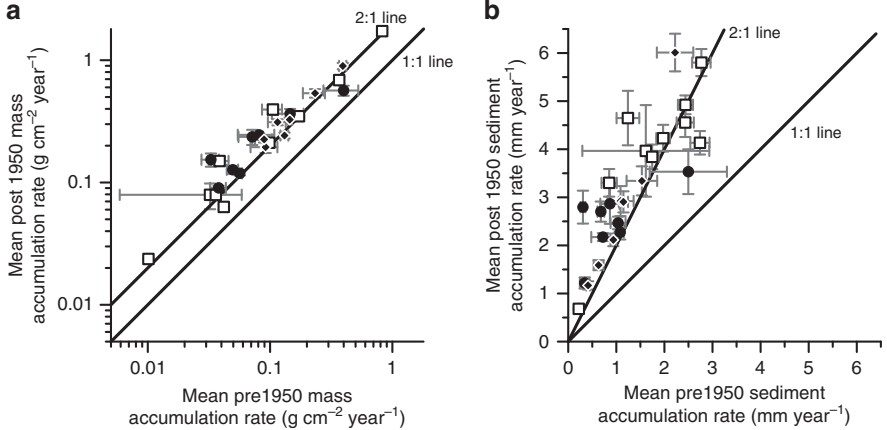

**Fig. 5 Mean pre and post 1950 mass accumulation rates and sediment accumulation rates.** Mean mass accumulation rates (± mean measurement error; **a**) and sediment accumulation rates (± mean measurement error; **b**) increased after 1950 at each site except for Sabine Bay. Notice mean mass accumulation rates are plotted using a logarithmic scale. Square = East Coast, FL, and AL (N = 10). Circle = Louisiana and Texas (N = 8). Diamond = Cuba, MEX, and West Coast (N = 7). Data provided in Supplementary Data 1.

sediment flux to the depocenters. Dams intercept a large percentage of the suspended sediment load from the watersheds up-river; however, supplemental sources of fine-grained alluvial sediments down-river of dam emplacement sites have been widely documented resulting from river incision and bank erosion[37–39]. Additional primary sediment sources in lower rivers from urbanization[40,41], agricultural development[42], and changes in riparian vegetation[43] have also been observed in shoreline-proximal areas. The transport of these additional fine-grained sediment sources to coastal depocenters is highly non-linear in time and space[44] and may involve lag times from decades to centuries and net transport velocities of <10 m yr⁻¹ [45]. Pulses in fine-grained sediment transport[46] and ephemeral discharges from tributaries[47] have also been observed in response to increased precipitation in the lower watershed and the mobilization of legacy sediments[48]. The post 1950 population growth in coastal counties is an indicator of degradation[49] and is commonly associated with increased runoff from land development and construction[50] and increased erosion

from land-cover changes[51] in shoreline-proximal areas. The sediments from eroding shorelines and degrading intertidal habitats are secondary sources to coastal depocenters, but for most of the sites this source is too low to explain the increase in MAR given the disproportionately large area of the depocenter relative to the area of shoreline loss (orders of magnitude difference) and that shoreline erosion increases the size of the depocenter. The exception being the coastal depocenters of Louisiana where these sources promoted the MAR increase post 1950 despite the distinct reduction in Mississippi River flow and suspended-sediment flux to the margin[52], which contributed to the loss of wetlands[53]. Rapid permafrost decay in river catchments from warming climate is a source of sediment to high latitude depocenters and contributed to the MAR increase in Nastapoka Sound[31]. This study falls short of identifying the relative additional contributions of specific downstream sediment sources to coastal depocenters; a necessary next step because each sediment source has a different implication for coastal-hazard mitigation, the carbon cycle, and the transport of contaminants attached to fine-grained particles.

**Sediment accumulation rates**. Measurements of SAR resolve changes in the thickness of the sedimentary record through time and is an important component of bathymetric change. At all sites, average SAR for the period since 1950 was more than double (within error) what it was for the period before 1950, (Fig. 5b). The SAR of many depositional environments have been shown to increase systemically with decreasing measurement duration in large data sets that include a wide range of time scales over which SAR was measured (millions of years to minutes)[54,55]. This relationship is referred to as the Sadler effect[55] and is mainly driven by the inclusion of larger hiatuses in the sedimentary record when measurements of SAR incorporate longer averaging time. It is unlikely that the measured increase in average SAR post 1950 is the result of more erosional and/or non-depositional periods in the pre1950 section of the ~100-year sedimentary record because the averaging time among individual measurements only ranges between years and decades. In addition, no recognizable differences in sediment composition with increasing SAR averaging time were documented in the cores. At 13 of our sites we tested the Sadler-effect prediction that average SAR measured over century and millennial time scales would be one- and two-orders of magnitude less, respectively, than SAR measured over decadal time scales (Supplementary Fig. 1, Supplementary Table 1, and Supplementary Data 3). At those sites, we compared SAR derived from published radiocarbon dates that provide averaging times ranging from 710 to 8380 yrs with measurements of SAR averaged over multiple decades. We used the part of the [210]Pb record older than 1890 in this comparison to minimize influence of the acceleration in SLR and related sediment accommodation that occurred between ~1880 and 1930 on SAR[7,56]. The average SAR measured using [210]Pb over decadal time scales was ≤ SAR measured over century to millennial time scales (within error) indicating that depositional and erosional processes are relatively constant in those depocenters during accumulation of the sedimentary record and that average post 1950 SARs are unprecedented.

**Sediment accommodation and relative sea level**. The increase in SAR over the last century must be associated with an increase in sediment accommodation across the North American coast, which at these multidecadal time scales would be driven by SLR and possibly a decrease in wave energy of the basin. It is unlikely that wave energy decreased or that pre 1950 sediments experienced more erosional events than post 1950 sediments because no evidence exists of a systematic decrease in storminess or estuarine

fetch across the North American continent during the twentieth century, rather, studies show an increase in large storms[57,58] and subtidal area[59], and no changes in sedimentology (texture, bedding, and bioturbation) were observed at that time horizon (Supplementary Note 1).

Global SLR must have contributed to the increase in accommodation[7,56], and global SLR affected each site differently over the last 50 years due to spatial variability in land movement, meteorology, and climate change[60]. For example, from 1950 to 2016 sea level near Nastopoka Sound, Canada fell 9.92 mm yr[−1] due to isostatic rebound, but in Barataria Bay, LA, sea level rose 9.40 mm yr[−1], three times the rate of global SLR due to subsidence (Supplementary Data 1). Correspondingly, average SAR post 1950 is lowest at the Nastopoka Sound site (0.54 mm yr[−1]). Except for Nastopoka Sound and the Pacific Northwest margin where sea level has been falling, both the rate of relative SLR[7,56] and SAR was higher at the sites during the post 1950 period than the preceding years.

During the period 1950-present, SAR at sites along the East and West coasts, FL, AL, Cuba, and Mexico matched or exceeded average rates of relative SLR, indicating static water depths or shallowing (Fig. 6). These sites are not sediment starved, rather, they are either in equilibrium with relative SLR or have a surplus of sediment. In contrast, average rates of relative SLR at the 8 sites in LA and TX far exceeded SAR over the same time, signifying a deepening of water and increasing subtidal area. Even though SAR increased after 1950 in TX and LA, sediment sources could not keep pace with rates of relative SLR, which is high due to natural (sediment loading and faulting) and anthropogenic (fluid withdrawal) processes[61]. In Barataria Bay, the rate of wetland loss increased from 0.36% yr[−1] during the period 1945–1956 to 1.03% yr[−1] during the period 1956–1969[62], like the broader Louisiana coast that experienced accelerated land loss around 1950[63]. Successful restoration and conservation of lost intertidal habitats in TX and LA will require supplemental sediment to maintain an optimal position in the tidal frame.

River impoundments are ubiquitous worldwide, with an estimated 16.7 million reservoirs > 100 m[2] (~8070 km[3] total

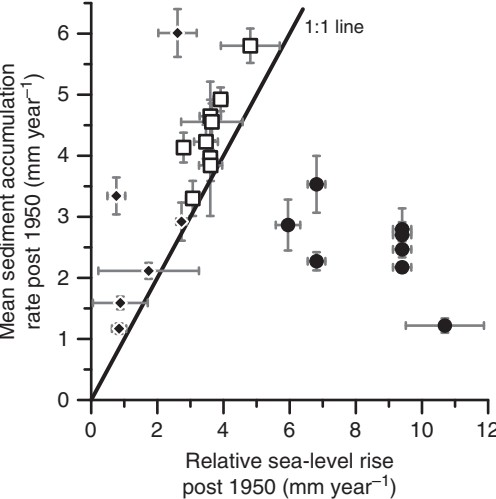

**Fig. 6 Water depth stable or shallowing at sites excluding TX and LA.** Mean sediment accumulation rates post 1950 (± mean measurement error) matches or exceeds average rates of relative sea-level rise (± s.d.) except for sites in Louisiana and Texas. Notice the exclusion of Nastapoka Sound and Pacific Northwest Margin sites where relative sea level is falling. Square = East Coast, FL, and AL (N = 9). Circle = Louisiana and Texas (N = 8). Diamond = Cuba, MEX, and West Coast (N = 6). Data provided in Supplementary Data 1.

volume), most of them constructed since 1950, creating inland accommodation and trapping sediment[64]. The specific timing for the rapid expansion of dam-construction projects varies among countries, is linked with human population growth and economic development[65], and the associated reduction in riverine sediment load downstream is thought to cause a deficit in coastal-sediment budgets globally[66]. This is clearly happening along the shorelines of deltaic coasts that are rapidly eroding[16] but may not be the case in subtidal settings of coastal depocenters or ~90% of deltas, worldwide[67]. Our work suggests that coastal depocenters away from deltaic headlands receive most of their sediment from coastal areas downstream of dams where anthropogenic land-cover changes are increasing and producing additional sediments. These sediment sources have more than compensated for sedimentation in upstream reservoirs and maintained static or decreasing water depths with relative SLR. Rapidly subsiding areas are an exception, and subtidal water depths will continue to increase there, despite increasing sedimentation. The regional empirical dataset presented here includes highly diverse settings, but nevertheless shows a coherent pattern of increased sedimentation in coastal depocenters challenging long-held assumptions that humans have decreased sediment supply to all coastal areas. The increase in sedimentation of 25 North American coastal depocenters is not unique to this continent and is likely manifested asynchronously across other regions given variations in the timing of human pressures on watersheds and societal responses[68–70]. While these data suggest coastal bathymetry is more resilient to SLR than previously thought, it also adds to the growing number of studies warning against future higher rates of SLR[61] shifting the balance between sediment accumulation and accommodation towards increasing water depths, globally, like what is happening now in Louisiana and Texas.

## Methods

**Data collection and analysis.** The findings of this study are based on a compilation of data, including original data from Louisiana, USA, but mostly from other published works (Supplementary References 1–9 and 12–23). Original data are from Terrebonne Bay, Barataria Bay, and the Louisiana inner continental shelf at 20 and 30 m water depth. At each site, we collected one core (10-cm diameter) using a gravity-coring device with a one-way valve to retain the sediments. We subsampled each core into 1.0-cm intervals and measured porosity and dry bulk density on each subsample from weight lost after freeze drying. The cores were composed of homogeneous silt and clay. X-radiographs revealed intact primary structures undisturbed by bioturbation. No discernable downcore $^{137}$Cs peaks were observed in these cores, consistent with previous studies, which conclude that the $^{137}$Cs supply for this region is derived primarily from the watershed and not from direct atmospheric flux[71]. We measured the total $^{210}$Pb activity of each subsample by gamma spectrometry (46.5-keV gamma peak)[28] and gamma counting was conducted using Germanium detectors (Coaxial-, BEGe-, and Well-types) coupled with a multi-channel analyzer. To calculate the $^{210}$Pb excess values, a background $^{210}$Pb activity was subtracted from total $^{210}$Pb activity. The background levels of $^{210}$Pb were determined by measuring the average gamma activities of $^{214}$Pb (at 295 and 352 keV) and $^{214}$Bi (609 keV).

**Site-selection criteria and geochronologies.** The study sites and associated $^{210}$Pb profiles, in addition to the sites in Louisiana addressed above, were selected based on the following criteria: (1) Data must be published in a peer-reviewed article that discusses sedimentation in coastal depocenters; (2) the core sites were away from human disturbances such as dredging or groin construction; (3) the $^{210}$Pb profiles have at least three data points before and three data points after 1950; and (4) the profiles have complete excess $^{210}$Pb inventories. Additionally, sites were selected to maximize the diversity of North American geologic and climatic settings included in the study. Brief core descriptions are provided in Supplementary Note 1.

Multiple cores were collected and reported on at 16 of the study sites and when multiple cores met our criteria, we chose to include the core with the most complete sedimentary record (minimal surface mixed layer and consistent lithology) in our study. It is typical of coastal depositional environments to have spatial variability in the fidelity of sedimentary records because processes vary across small spatial scales, such as the heads of estuaries being river dominated and the mouths of estuaries being marine dominated. Importantly, the sedimentology and geochemistry of the cores, as presented in the publications from which we

extracted the information, demonstrate that the depth of surface mixing, bioturbation, and texture were relatively constant over the past 150 years. Information from X-radiographs of the cores were presented in 10 of the studies to evaluate mixing.

Once $^{210}$Pb profiles were identified that met the above criteria, data were extracted from publications by copying tables ($N = 10$) or digitizing graphs with Didger® ($N = 11$). Digitizing error was included along with the extraction of error bars from the graphs. Data, including excess and total $^{210}$Pb, dry bulk density (DBD) and/or porosity, were transferred into spreadsheets[30] for geochronological modeling. To account for changes in porosity with depth, some articles reported dry bulk density for each interval, but most reported an average DBD or porosity. A whole-core average value of DBD was used in our study to promote consistency across study sites. When DBD was not reported, porosity was used to calculate DBD and when neither were reported, porosity from a separate peer-reviewed article in the same or nearby core location was used. To support the reported DBD values and evaluate changes in depositional processes, 16 and 8 of the studies included grain-size and % carbon information, respectively. We used Equation 1 to calculate DBD from porosity, where 2.65 g cm$^{-3}$ is the average density of continental crust material.

$$\text{DBD} = 2.65\,(1 - \text{porosity}) \tag{1}$$

The median core length was 44 cm (max. 346 cm and min. 18 cm) and the constant rate of supply or constant flux[30] model was used to obtain rates of sediment accretion (SAR) and mass accumulation (MAR) for every subsampled interval because it allows sedimentation rates to vary over time. This method yielded dates to about 1880 and the median depth to supported $^{210}$Pb activity was 35 cm (max. 170 cm and min. 7 cm). Each value of SAR and MAR has a measurement error associated with it. SAR uncertainty is the square root of the sum of the squares of 4 values, including: (1) decay constant uncertainty/decay constant, (2) inventory below the unit uncertainty/inventory below, (3) activity uncertainty/activity, and (4) dry bulk density uncertainty/dry bulk density[30]. In addition to $^{210}$Pb profiles, 20 of the studies included information on bomb fallout nuclides $^{137}$Cs and/or $^{239, 240}$Pu to validate the average SAR and MAR of the core and evaluate mixing. To test the null hypothesis that the average of the differences between the pre and post 1950 MAR and SAR values is zero, we used a paired $t$-test. Means and $p$-values are provided in Supplementary Data 1.

The Santa Clara Shelf (Site 23) and Pacific Northwest Margin (Site 25) are located on the outer continental shelf where the potential for an additional source of excess $^{210}$Pb from upwelled waters exists. Upwelled waters could bring a variable $^{210}$Pb source (higher or lower) than supplied by atmospheric deposition and therefore violate the assumption of a constant flux of $^{210}$Pb to the site, a requirement for the constant rate of supply model. How upwelling might compromise the geochronology model would depend on (a) the concentration of $^{210}$Pb in the upwelled waters; and (b) the frequency of upwelling, but no information exists regarding $^{210}$Pb concentrations in upwelled waters at the two sites. At both sites upwelling is seasonal occurring during the summer months[72,73]. Given that cores were subsampled at 1-cm intervals and sites 23 and 25 have average SAR values of 0.8 and 2.1 mm yr$^{-1}$, respectively, each sampling interval integrated over multiple upwelling cycles. The frequency of variability for $^{210}$Pb supply from upwelling to each site would be less than the minimum time interval represented by each subsample and should therefore have minimal impact on the centimeter by centimeter geochronologies established.

**Population.** Population was queried at each site using hydrologic units that bordered the depocenter. Population shapefiles and tables for the United States were obtained from National Historical Geographic Information System[74] at the county level for 1900, 1950, and 2010. Watershed shapefiles were collected from the United States Geological Survey https://water.usgs.gov/GIS/huc.html. Population was integrated for each county within hydrologic unit 10. Population data for areas falling outside of the US were obtained from the literature[75–77] or from a government website for times nearest 1900, 1950, and 2010 (see Supplementary Data 1).

**Sea level.** We obtained annual mean sea-level data from tide gauges closest to each site using the Permanent Service for Mean Sea Level (PSMSL), 2018, "Tide Gauge Data", retrieved June 2018 from http://www.psmsl.org/data/obtaining/ (see Supplementary Data 1)[78]. To obtain average rates of sea-level change and error, we applied a linear regression to those post 1950 data. Not all gauges were in operation over the entire 1950 to present period (see Supplementary Data 1 for details).

## Data availability
The authors declare that the data supporting the findings of this study are available in Supplementary Note 1, Supplementary Fig. 1 and Supplementary Data 1–3.

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

## Acknowledgements

This research was supported by the North Carolina Policy Collaboratory and The University of North Carolina at Chapel Hill Graduate School.

## Author contributions

A.B.R. wrote the paper, drafted the figures, and processed the sea-level data. B.A.M. selected the sites, provided data, and modeled geochronologies. C.B.M. gathered population data. A.B.R., B.A.M., C.B.M., M.C.B., and A.N.A. participated in study conception, compiling the dataset, QA/QC of results, and editing the paper.

## Competing interests

The authors declare no competing interests.
