## [Peer Review File · Nature Communications]

Reviewers' comments:

Reviewer #1 (Remarks to the Author):

The authors propose to test the assumption that the decrease in river sediment loads as a consequence of dam construction is propagated to the coastline, resulting in a hypothetical reduction in suspended sediment delivery to coastal depocenters. This is of global importance because rising sea level is creating accommodation faster than sediment supply can potentially fill it. This could lead to loss of intertidal and subaerial land, habitat transformation, and landscape alteration.

I concur with the authors that the assumption that there has been a reduction in sediment supply to coastal watersheds is urgently in need of testing. The authors have used the most abundant type of sedimentation rate measurement, Pb-210 geochronology, and have utilized a conservative approach to evaluating previously published studies. I also have made similar type measurements in this depositional environment. Establishing sedimentation rates from these data can be challenging for a number of reasons that the authors explain. In general, I like the rationale for the manuscript, the data analysis approach, and time expended to assemble these data for analysis. Yet, there are a few issues that they need to expand on further in my opinion before this is ready for publication. I will address these in detail, and some may preclude its publication in Nature Communications.

A simple recommendation is to change the title to "Coastal sedimentation in North America doubled during the Anthropocene despite river dams" because only studies from North America are included. There are dozens of additional studies that I am aware of that used Pb-210 geochronology to examine coastal sedimentation (e.g., Greg Brunskill's Pb-210 studies in Australia and New Zealand to start with). I appreciate that assembling these data are time consuming, which may have been a reason that only North American studies were included. However, for a global journal such as Nature Communications, the authors of this manuscript should include a paragraph or two explaining why only NA sites were used and how the potential results of this NA-centric study might apply globally.

There are many studies that I am aware of that use Pb-210 geochronology in coastal N. American watersheds that were not included here. The authors should include a supplementary file that includes ALL the studies that were examined for use but were subsequently excluded for the criteria used here to select only the most robust data sets for study. This is important because it will elevate any perception that the authors singled out data sets that only supported their hypothesis. The conclusions of this manuscript would be seen as more comprehensive if they report that these other studies were examined but not used, explicitly stating which criteria were violated for each.

I concur with the authors statement starting on line 281 that "It is typical of coastal depositional environments to have spatial variability in the fidelity of sedimentary records because processes vary across small spatial scales, such as the heads of estuaries being river dominated and the mouths of estuaries being marine dominated." For example, we know from decades of estuarine studies that fine-grain sediment accumulation is focused in turbidity maxima. However, I can see a major criticism of this study would be that only ONE sediment core was used for each geographic region to demonstrate the sediment supply at large hasn't decreased due to dam construction. What if all the other cores in the estuary showed no change (or a decrease) in sedimentation rates? Does that mean that these locations are sediment starved in general or does it mean that the one site chosen is just in a unique hydrodynamic setting that allows for focused deposition? Also, the authors need to directly address the Sadler effect, which is a demonstrated apparent increase in sedimentation rates as the time scales of investigation decrease (e.g., (1) Sadler, P. M., and D. J. Jerolmack, 2015: Scaling laws for aggradation, denudation and progradation rates: the case for time-scale invariance at sediment sources and sinks. Geol. Soc. London, Spec. Publ.,

404, 69–88, <https://doi.org/10.1144/SP404.7>. (2) Schumer, R., and D. J. Jerolmack, 2009: Real and apparent changes in sediment deposition rates through time. *J. Geophys. Res. Solid Earth*, 114, F00A06, <https://doi.org/10.1029/2009JF001266>.). Ideally, if you want to show that there has been an increase in sedimentation rates since some temporal datum, one should be working with sediment volumes, not rates. But this requires spatial data (e.g., seismic reflection profiles; bathymetric changes) for each estuary, which are harder to acquire than sediment cores. If nothing else, the authors of this paper will need to address the Sadler effect as a potential influence on their results and how using just a single sediment core for each location can really be justified given what we know about sediment focusing for most estuaries.

I would recommend not including data from studies that examine continental shelf sedimentation. Foremost, changes in sediment delivery to a location on a continental shelf may have nothing to do anthropogenic alteration of sediment yields. For example, an increase in sedimentation in a continental shelf core since the mid 20th century could be due to increases in flood events that supply more sediment (e.g., Eel River shelf, northern California; Sommerfield et al., *Geology*, 2002: [/10.1130/0091-7613\(2002\)030<0395:SROCCI>2.0.CO;2](https://doi.org/10.1130/0091-7613(2002)030<0395:SROCCI>2.0.CO;2).) Trying to unravel why sedimentation rates on a north American continental shelf have changed is beyond the scope of a paper like this that is designed to test the assumption that dam construction in watersheds has reduced the sediment supply to coastlines.

I appreciate the authors inclusion of uncertainty in their results (i.e., Figure 3). However, there is no explanation of how it was specifically calculated. Lines 302-303 simply state “Each value of SAR and MAR has a measurement error associated with it.” The reader should know how it was calculated. Also, the authors may not be aware, but there is a new Bayesian approach for uncertainty analyses using the CRS model for Pb-210 geochronology:

Aquino-López, M. A., M. Blaauw, J. A. Christen, and N. K. Sanderson, 2018: Bayesian Analysis of 210Pb Dating. *J. Agric. Biol. Environ. Stat.*, 1–17, <https://doi.org/10.1007/s13253-018-0328-7>.

Martin Blaauw created the “Bacon” Bayesian modelling approach for C-14 data that most everyone now uses (and should!) and I have begun to use this new approach for Pb-210 dating using the CRS model. I encourage the authors to explore its use for this project.

Sincerely,
John M. Jaeger

Reviewer #2 (Remarks to the Author):

I have reviewed “coastal sedimentation doubled during the Anthropocene despite river dams”, where Rodriguez et al compile sediment core accumulation rate studies to describe recent changes to sediment rates across the US and Canada. Their main conclusion is that sediment cores show an increase in sedimentation rates, contrasting an expected decline of the sediment flux to the coastal ocean from river dam construction.

I find the data compilation to be very interesting and potentially suitable for publication in *Nature Communications*. However, even though the conclusion is relatively straightforward, I find the manuscript to be organized rather confusingly. Several statements are repeated multiple times in different sections, whereas other aspects are glossed over quickly. It is still readable because the story is not very complex, but nevertheless it could be improved a lot.

1. For example, the aim of the study is not put forth until L96. The text before L96 is a mix of background/introduction/motivation.

2. L90 in the middle of the section suddenly brings up a hypothesis in what appears to be a background section.
3. Next, in the middle of a section (L135) the authors begin to describe their results. After this initial result, however, the authors start to discuss methods again (L141).
4. Throughout the text, it remains unclear what exactly MAR and SAR represent, what units these two variables have, and why sediment accumulation needs two separate variables. What is the difference between "sediment mass accumulation rate" and "sediment accumulation rate", the latter is a volumetric rate?
5. L105 seems to be a methods section, but only includes an in-depth discussion of 210Pb/geochronology. Other relevant data such as RSLR, dams, etc are mentioned in the separate methods section.
6. Relative SLR is mentioned frequently as creating the accommodation space needed to produce the observed increase in MAR (or SAR). However, rather than having a dedicated section on RSLR and an in depth analysis, RSLR is mentioned from L168 to L226 as (1) contributing to accommodation, (2) reducing accommodation (Nastopoka), (3) lagging behind RSLR (LA, TX), (4) not explaining the increase in accommodation, and (5) exceeding sedimentation. I am not saying these statements are necessarily incorrect, but simply that the order and analysis makes for a unnecessarily confusing read. I would suggest adding a dedicated figure or analysis comparing SAR/MAR to RSLR (which can be written to have the same units).

Aside from the presentation, I have a couple of significant concerns about the data itself.

7. The main conclusion, namely that alternative sediment sources downstream of dams are responsible for increased coastal sedimentation, is not verified independently. The authors should try to find examples of such sediment sources, or mention very clearly (in the abstract preferably) that they were not able to find any evidence.
8. Analyses of changes to sedimentation rates are susceptible to a bias known as the Sadler effect (e.g. [dx.doi.org/10.1029/2009JF001266](https://doi.org/10.1029/2009JF001266)), which is an apparent (but not real) increase in sedimentation rates in time resulting from erosion. It appears to me that this bias might also apply to the data presented by the authors. Can they correct for this?, or perhaps explain in the methods section why it does not apply to their record?

Minor comments

L2 I would suggest to write '1950' instead of 'Anthropocene' because the authors specifically refer to 1950, and the term anthropocene is ambiguous.

The abstract is rather confusingly vague. The first sentence is too long, L19: upstream from where, L20 'total sediment supply' is nonspecific, L23 MAR and SAR should be more clearly defined, L24 I would say that a recent increase in land degradation would also be able to explain increased sedimentation, L27 'lagging 2-8 behind' rewrite to make it less ambiguous.

L47 'some': give examples (CRMS studies in LA seem to be focused on sedimentation)

L77 '8-fold' depends on the number of dams prior to 1950, I would be more specific

L105 "researchers", who?

L151 but there is also great land loss in other nearby places in the gulf? In general I would encourage the authors to discuss their results in the context of land loss reported in many coastal places

L200: reference?

L212: reference 60 (church and white) does not really validate this statement I believe? Is there a statistical test the authors can refer to which shows that variability in RSLR does not explain the observed MAR/SAR trends

L231: I am unclear what exactly the supposed difference is between coastal depocenters and deltaic coasts.

L252: "below"? in general the methods section is also rather unclear and could use better subdivision into separate section.

L171 I would say that an increase in wave energy would lead to enhanced accommodation space. I

am probably mistaken because I find the term accommodation space a little confusing, but I would think that increasing waves would lead to more efficient export of sediment from the coastal zone, and hence lead to more space for sediment to be deposited?

In conclusion, I think the topic is of great interest to many researchers. I find the manuscript however to be more complicated than it needs to be. I think that greater emphasis on coastal processes that lead to sediment transport and sediment erosion/deposition (waves, tides, rivers) and better subdivision of the article into sections would help the study significantly.

We received two reviews of the manuscript. Comments, edits, and suggestions are pasted below in italics, with our responses in plain text. We thank both reviewers for the time they spent reading the paper and their generosity with sharing ideas.

Reviewer #1 Dr. John M. Jaeger:

1. A simple recommendation is to change the title to “Coastal sedimentation in North America doubled during the Anthropocene despite river dams” because only studies from North America are included. There are dozens of additional studies that I am aware of that used Pb-210 geochronology to examine coastal sedimentation (e.g., Greg Brunskill’s Pb-210 studies in Australia and New Zealand to start with). I appreciate that assembling these data are time consuming, which may have been a reason that only North American studies were included. However, for a global journal such as Nature Communications, the authors of this manuscript should include a paragraph or two explaining why only NA sites were used and how the potential results of this NA-centric study might apply globally.

We changed the title to include North America. Yes, we agree that there are dozens of additional studies that used Pb-210 geochronology to examine coastal sedimentation around the world. In North America, the largest increase in dams and coastal population occurred after the second world war but the timing of human pressures on watersheds and societal responses varies globally. River impoundments are related to the economic and population growth of countries which increases asynchronously. We excluded areas outside of North America because Australia, New Zealand, China, the UK and other locations have a different history of development and modification of river networks than North America, which would complicate our grouping of the sedimentary record into pre- and post-1950 time periods. We have addressed this by reorganizing the manuscript. The first paragraph of the “Results and discussion” section now presents that the timing of dam construction and coastal population increase is specific to North America. Figure 4, which presents information about coastal populations, is now Figure 2. In the last paragraph of the manuscript we added these two sentences: “The specific timing for the rapid expansion of dam-construction projects varies among countries, is linked with human population growth and economic development (Zarfl et al., 2015), and the associated reduction in riverine sediment load, downstream, is thought to cause a deficit in coastal-sediment budgets globally (Maybeck, 2003)” and “The increase in sedimentation compiled here for 25 coastal depocenters of North America are not unique to this continent and are likely manifested asynchronously across other regions given variations in the timing of human pressures on watersheds and societal responses (Emeis et al., 2000; Oldfield et al., 2003; Owen and Lee, 2004; Lu and Matsumoto, 2005).”

2. There are many studies that I am aware of that use Pb-210 geochronology in coastal N. American watersheds that were not included here. The authors should include a supplementary file that includes ALL the studies that were examined for use but were subsequently excluded for the criteria used here to select only the most robust data sets for study. This is important because it will elevate any perception that the authors singled out data sets that only supported their

hypothesis. The conclusions of this manuscript would be seen as more comprehensive if they report that these other studies were examined but not used, explicitly stating which criteria were violated for each.

We never intended to suggest that our data set is comprehensive of all Pb-210 geochronologies in coastal N. America. It's worth noting that many sites we have studied were not included in this compilation, simply because we had other sites nearby, not because they didn't show a post 1950 increase in MAR and SAR (e.g. Mattheus et al., 2009). We agree that it seems like a good idea to generate a list of all papers and chronologies we looked at and rejected, but we feel generating and publishing a list of papers we believe have shortcomings is unconventional. We respectfully disagree that there will be any perception that we singled out these 25 sites around North America because they all supported our hypothesis. There is no motivation for this, especially considering this study was mainly conducted during a small seminar class. Please see our response to comment 3 below, which should ease any concern that we cherry picked the data set.

3. I concur with the authors statement starting on line 281 that “It is typical of coastal depositional environments to have spatial variability in the fidelity of sedimentary records because processes vary across small spatial scales, such as the heads of estuaries being river dominated and the mouths of estuaries being marine dominated.” For example, we know from decades of estuarine studies that fine-grain sediment accumulation is focused in turbidity maxima. However, I can see a major criticism of this study would be that only ONE sediment core was used for each geographic region to demonstrate the sediment supply at large hasn't decreased due to dam construction. What if all the other cores in the estuary showed no change (or a decrease) in sedimentation rates? Does that mean that these locations are sediment starved in general or does it mean that the one site chosen is just in a unique hydrodynamic setting that allows for focused deposition?

We certainly are only reporting results from one sediment core from each depocenter and agree that a common criticism of studies that present information from one core is “how representative is that core of the entire depositional environment?” We didn't clearly communicate in supplementary information that most of these studies included multiple cores from the depocenter. This information is now stated in the Methods section, “Multiple cores were collected and reported on at 16 of the study sites...” In supplementary information we also summarize SAR and MAR reported for all cores collected at the sites and present our reasoning for selecting the core we did. All changes to supplementary information are highlighted in the file. It is worth noting that for most sites, we did not choose the record with the highest SAR and MAR (the site that maximizes sediment focusing and corresponds with the turbidity maximum). What is key is that those erosional and depositional processes that build the sedimentary record at the core site are constant through time. This can be evaluated somewhat by the Pb-210 profiles, X-radiographs, and impulse tracers. Yes, many cores included in our study were reported to show no change in sedimentation rate through time, but that is because the authors used the “Constant Initial Concentration” dating model, which assumes sedimentation rates are

constant through time. This is addressed in the paper at the second paragraph of the “Results and discussion” section and in Methods. The authors of the papers that address sedimentation in the Pettaquamscutt River Basin, Pokomoke Sound, Pamlico River Estuary, Neuse River Estuary, and Cienfuegos Bay recognized and reported a post 1950 increase in sedimentation and those studies were based on multiple cores. That is all now presented clearly in supplemental information.

4. Reviewer 2 had a similar comment and we address both concerns here. Also, the authors need to directly address the Sadler effect, which is a demonstrated apparent increase in sedimentation rates as the time scales of investigation decrease (e.g., (1) Sadler, P. M., and D. J. Jerolmack, 2015: Scaling laws for aggradation, denudation and progradation rates: the case for time-scale invariance at sediment sources and sinks. Geol. Soc. London, Spec. Publ., 404, 69–88, <https://doi.org/10.1144/SP404.7>. (2) Schumer, R., and D. J. Jerolmack, 2009: Real and apparent changes in sediment deposition rates through time. J. Geophys. Res. Solid Earth, 114, F00A06, <https://doi.org/10.1029/2009JF001266>.) Ideally, if you want to show that there has been an increase in sedimentation rates since some temporal datum, one should be working with sediment volumes, not rates. But this requires spatial data (e.g., seismic reflection profiles; bathymetric changes) for each estuary, which are harder to acquire than sediment cores. If nothing else, the authors of this paper will need to address the Sadler effect as a potential influence on their results and how using just a single sediment core for each location can really be justified given what we know about sediment focusing for most estuaries.

The Sadler effect is an apparent decrease in SAR with increasing time over which SAR is measured and is due to the inclusion of longer hiatuses in deposition (or larger erosional events) as the averaging time increases. Basically, SAR measured over longer time scales includes more periods of non deposition than SAR measured over short timescales which results in an apparent decrease in SAR as the measurement time interval increases. Data used to support the Sadler effect is shown on a LOG-LOG scale with accumulation rate on the Y axis (fraction of a mm/yr to many m/year) and time interval on the X axis (minutes to many millions of years). Sadler Plots show that SAR is inversely related to the time scale over which it was determined and this is largely driven by including changing depositional environments, subaerial exposure surfaces, ravinement surfaces, avulsion events, climate change, sea-level change, etc. in the calculation of SAR, which can be addressed by looking at the sedimentology and stratigraphy of an area. All cores in our study are sampling a single depositional environment over a period of Earth’s history that is relatively well constrained in terms of sea level, storms, and landscapes. In addition, Sadler Plots include various dating techniques, do not present clear selection criteria to ensure data quality, and lump together important depositional and erosional processes that operate over various time scales. Our study only relies on Pb-210, which is best suited for measuring changes in SAR because it follows the pathway of clay and silt particles. The “Sadler effect” is real, but in our opinion the most important lesson from the numerous papers written on the topic is that it can be avoided. This is important to demonstrate in a study such as ours.

So how does the Sadler Effect apply to our study? We compiled Pb-210 profiles from the literature mainly from the middle part of estuaries. Each profile is based on a core that was

sectioned continuously at some interval and we have a date for each interval of the core (we are not integrating between the top of the core and a date at some depth like a Sadler Plot). As we move down the core into older sediments, SAR decreases and the time interval each core section represents increases at all our sites. The deepest slice of the core could represent 20 years of deposition, while the shallowest slice of the core could represent 2 years of deposition. Is this because the supply of sediment to the area and accommodation space is changing (our conclusion) or because the deepest 1-cm thick slice of the core includes more periods of nondeposition than the top 1 cm thick slice of the core and we are measuring an apparent increase in MAR and SAR post 1950? The sedimentological expression of increased erosion or nondeposition in coastal environments is commonly a change in lithology and/or sedimentary structures. More periods of erosion might manifest itself as coarser sediment, reflecting the greater energy expended on the bed. More periods of non-deposition might manifest itself as an increase in bioturbation as bottom-dwelling organisms have more time to spend on the bed. We present in supplemental materials that these cores are basically homogeneous in terms of lithology and sedimentary structures, which would negate the inclusion of additional periods of erosion and/or nondeposition in the deepest section versus the shallowest section. We neither include the surface mixed layer in our calculations, where the Sadler effect is truly impacting deposition, nor cross contacts between depositional environments at depth in the cores, which would manifest itself as a transgressive or regressive surface and a change in sedimentary processes.

Let's assume that additional periods of erosion and/or nondeposition in the deepest section of the core simply cannot be resolved with grain-size data, X-rays, or any of the other parameters the original authors measured down core. Another way to test the Sadler effect is to extend the SAR record to the 1000-year time scale. If the Sadler Effect is influencing our sites, then SAR at the 1000-year time scale should be an order of magnitude lower than SAR measured at the base of our 150-year records. We found publications that present radiocarbon dates from greater core depths at 13 of our sites (see new supplementary table 3). We use the age and depth of the radiocarbon dates and the basal age and depth of the Pb-210 profiles to calculate SAR at the millennium time scale. The Pb-210-based SAR values for sections older than 1890 CE and pre 1950 are compared with SAR measured at the millennium time scale in supplementary figure 1 (at the end of the supplementary information file). This allows us to minimize the effect of increasing rates of sea-level rise between 1880 and 1930 on the analysis. Average millennium time-scale rates are higher or the same as those pre 1890 rates that are based on Pb-210 data showing that we have avoided the "Sadler effect". Average pre 1950 SAR values are higher, largely the result of an increase in sediment accommodation, but many are on and below the 1:1 line. Post 1950 SAR rates are truly unprecedented. This new data set and plot should address the Sadler Effect and places our results in context with the last few thousand years. We added new text to the manuscript that directly addresses the Sadler effect in the "Sediment accumulation rates" section.

For this study, imaging volumes with seismic data is impossible. Most of our records are based on the shallow sedimentary record that is less than 0.5 m thick. Many high-resolution seismic methods, such as chirp sonar, report a theoretical resolution at the decimeter scale, but in

practice that is almost never achieved. We report that the sediments sampled in the cores are homogeneous, so there is no reflecting interface associated with 1950. One would need some contrast in acoustic impedance across a boundary that corresponds to 1950 and the interface would have to extend throughout the area of the site allowing us to map volumes with seismic methods, but that boundary does not exist.

5. I would recommend not including data from studies that examine continental shelf sedimentation. Foremost, changes in sediment delivery to a location on a continental shelf may have nothing to do anthropogenic alteration of sediment yields. For example, an increase in sedimentation in a continental shelf core since the mid 20th century could be due to increases in flood events that supply more sediment (e.g., Eel River shelf, northern California; Sommerfield et al., Geology, 2002: /10.1130/0091-7613(2002)030<0395:SROCCI>2.0.CO;2.) Trying to unravel why sedimentation rates on a north American continental shelf have changed is beyond the scope of a paper like this that is designed to test the assumption that dam construction in watersheds has reduced the sediment supply to coastlines.

Yes, other factors like changes in weather can modify sedimentation, but even Sommerfield et al., 2002 recognized that human impacts contributed to the increase in sedimentation that they documented for the Eel River shelf as stated in the last sentence of their abstract, “Anthropogenic increase in watershed-sediment production is a probable secondary factor”. Our results are also consistent with conclusions from studies in other marine areas around the world such as the Gulf of Tehuantepec, Mexico (Ruiz-Fernandez et al., 2009; included in our North American survey), the Baltic Sea (Emeis et al., 2000), the Adriatic sea (Oldfield et al., 2003) and the pericontinental shelf of Hong Kong (Owen and Lee, 2004), where the observed increased input from land-sourced material reflects deforestation and expanding human impacts during periods of major land-use change.

Emeis, K.C. et al., 2000. Changes in the burial rates and C:N:P ratios in Baltic Sea sediments over the last 150 years. *Marine Geology* 167 (1–2), 43–59.

Oldfield, F. et al., 2003. A high resolution late Holocene palaeo environmental record from the central Adriatic Sea. *Quaternary Science Reviews* 22, 319–342.

Owen, R.B. and Lee, R., 2004. Human impacts on organic matter sedimentation in a proximal shelf setting, Hong Kong. *Continental Shelf Research* 24, 583–602.

*6. I appreciate the authors inclusion of uncertainty in their results (i.e., Figure 3). However, there is no explanation of how it was specifically calculated. Lines 302-303 simply state “Each value of SAR and MAR has a measurement error associated with it.” The reader should know how it was calculated. Also, the authors may not be aware, but there is a new Bayesian approach for uncertainty analyses using the CRS model for Pb-210 geochronology: Aquino-López, M. A., M. Blaauw, J. A. Christen, and N. K. Sanderson, 2018: Bayesian Analysis of 210Pb Dating. *J. Agric. Biol. Environ. Stat.*, 1–17, <https://doi.org/10.1007/s13253-018-0328-7>. Martin Blaauw*

created the “Bacon” Bayesian modelling approach for C-14 data that most everyone now uses (and should!) and I have begun to use this new approach for Pb-210 dating using the CRS model. I encourage the authors to explore its use for this project.

We now state how error was calculated in the methods section “Our calculation for SAR uncertainty utilizes the approach documented by Sanchez-Cabeza and Ruiz-Fenández (2012), which is the square root of the sum of the squares of 4 values, including: 1. decay constant uncertainty/decay constant, 2. inventory below the unit uncertainty/inventory below, 3. activity uncertainty/activity, and 4. dry bulk density uncertainty/dry bulk density.”

We are supportive of approaches to standardize Pb-210 usage for estimates of sediment accumulation rates and of efforts to produce realistic estimates of Pb-210 dating uncertainties. The Bayesian approach presented by Aquino-López et al. (2018) is a step in the right direction. To avoid some of the shortfalls in the calculation of uncertainty using the CRS approach we analyze each cm in the core down to where there is no longer excess Pb-210. Our criteria also required exclusion of any core data where the cores are not deep enough to reach background Pb-210 activities. Since we are, in most cases, using data from previously published work, we did encounter profiles where not every section down core were analyzed for Pb-210. In those cases, we interpolated between sections as suggested by Appleby (2001). Using our stated criteria eliminated using many previously published Pb-210 profiles. Similar lack of data prevents us from using the Bayesian approach to calculating rate uncertainties. In particular, the rate of supply of Pb-210 to the site and the counting error related to the equipment used to measure the samples are seldom given. Dr. Jaeger’s point is nevertheless well taken; there needs to be a standardization in the literature whereas all the relevant information used in calculating Pb-210 sedimentation rates is given. This would make efforts such as ours in this manuscript much more successful.

Reviewer #2:

I find the data compilation to be very interesting and potentially suitable for publication in Nature Communications. However, even though the conclusion is relatively straightforward, I find the manuscript to be organized rather confusingly. Several statements are repeated multiple times in different sections, whereas other aspects are glossed over quickly. It is still readable because the story is not very complex, but nevertheless it could be improved a lot.

This is an important criticism and we agree that the manuscript is improved after we reorganized it based on the comments listed below and eliminated redundancies. We also added section headings, using the guidelines provided by the journal, to help in this regard.

1. For example, the aim of the study is not put forth until L96. The text before L96 is a mix of background/introduction/motivation.

We now have an “Introduction” heading and state the aim of the study at the end of the introduction. We agree that the text went back and forth between background/introduction and motivation. This has been addressed by adding section headings and moving text around appropriately.

2. L90 in the middle of the section suddenly brings up a hypothesis in what appears to be a background section.

Yes, this statement was out of place and was moved to the end of the Introduction. The hypothesis is also part of a separate paragraph now.

3. Next, in the middle of a section (L135) the authors begin to describe their results. After this initial result, however, the authors start to discuss methods again (L141).

Those methods at L141 have been moved up in the text to the first paragraph of the “Results and discussion” section.

4. Throughout the text, it remains unclear what exactly MAR and SAR represent, what units these two variables have, and why sediment accumulation needs two separate variables. What is the difference between “sediment mass accumulation rate” and “sediment accumulation rate”, the latter is a volumetric rate?

In addition to units being included on all our figures, we now define the units of MAR and SAR in the abstract and at the beginning of the “Results and discussion” section (first sentence). Mass accumulation rates indicate changes in the mass of sediment accumulating through time ($\text{g cm}^{-2} \text{ yr}^{-1}$) and sediment accretion rates indicate how accommodation is being filled through time (cm yr^{-1}). Both are necessary. Our cores have little variation in composition, so increasing MAR indicates increasing flux of sediment to the site. Increasing SAR indicates increasing elevation of the bed and can be compared with relative sea-level rise.

5. L105 seems to be a methods section, but only includes an in-depth discussion of ^{210}Pb /geochronology. Other relevant data such as RSLR, dams, etc are mentioned in the separate methods section.

With the new organization of the paper, that is now the beginning of the “Results and discussion” section. We could include all this information in the Methods section, presented at the end of Nature Communications papers, but worry that interested readers will have difficulty following the manuscript without some general introduction to Pb-210 because those data are the crux of our entire study. The information about the impulse tracers Cs-137 and Pu-239, 240 was already addressed in the Methods section and that was deleted.

6. *Relative SLR is mentioned frequently as creating the accommodation space needed to produce the observed increase in MAR (or SAR). However, rather than having a dedicated section on RSLR and an in depth analysis, RSLR is mentioned from L168 to L226 as (1) contributing to accommodation, (2) reducing accommodation (Nastopoka), (3) lagging behind RSLR (LA, TX), (4) not explaining the increase in accommodation, and (5) exceeding sedimentation. I am not saying these statements are necessarily incorrect, but simply that the order and analysis makes for a unnecessarily confusing read. I would suggest adding a dedicated figure or analysis comparing SAR/MAR to RSLR (which can be written to have the same units).*

We now have dedicated sections “Sediment accommodation and relative sea level” and “Sediment flux to coastal depocenters”. Relative sea-level rise and SAR have the same units, but MAR has different units ($\text{g cm}^{-2} \text{ yr}^{-1}$). Figure 6 is a dedicated figure that compares SAR with RSLR. We have revised the text to minimize the back and forth between relative SLR and changes in accommodation.

7. *The main conclusion, namely that alternative sediment sources downstream of dams are responsible for increased coastal sedimentation, is not verified independently. The authors should try to find examples of such sediment sources, or mention very clearly (in the abstract preferably) that they were not able to find any evidence.*

We now have a new section “Sediment supply to coastal depocenters” and a brief statement in the abstract where we address specific sources of sediment. There are many examples of alternative sediment sources downstream of dams and we have included over 10 new references that independently verified those sources.

8. *Analyses of changes to sedimentation rates are susceptible to a bias known as the Sadler effect (e.g. [dx.doi.org/10.1029/2009JF001266](https://doi.org/10.1029/2009JF001266)), which is an apparent (but not real) increase in sedimentation rates in time resulting from erosion. It appears to me that this bias might also apply to the data presented by the authors. Can they correct for this?, or perhaps explain in the methods section why it does not apply to their record?*

This comment is addressed above in the response to Reviewer 1’s comment 4.

Minor comments:

L2 I would suggest to write ‘1950’ instead of ‘Anthropocene’ because the authors specifically refer to 1950, and the term anthropocene is ambiguous.

We changed the title to be more specific. It now reads, “Coastal sedimentation across North America doubled in the 20th century despite river dams “

The abstract is rather confusingly vague. The first sentence is too long, L19: upstream from where, L20 'total sediment supply' is nonspecific, L23 MAR and SAR should be more clearly defined, L24 I would say that a recent increase in land degradation would also be able to explain increased sedimentation, L27 'lagging 2-8 behind' rewrite to make it less ambiguous.

The abstract was reduced in length and is now close to the 150 word limit. The first sentence was shortened by deleting “Rivers are the principal contributors of sediment to coastal areas”. We meant upstream from the coast and this has been added. We deleted “total sediment supply.” We included the units of SAR and MAR to clarify what that means. We list some of the sources of sediment downstream of dams including “human-modified landscapes.” We modified the sentence that begins on L27 to clarify “...lagging 2-8 times behind”.

L47 'some': give examples (CRMS studies in LA seem to be focused on sedimentation)

We modified the text and include two examples, Day et al., 1999 and Morris et al., 2002.

L77 '8-fold' depends on the number of dams prior to 1950, I would be more specific

Yes, we agree the statement is vague. The text now reads, “...increased from 16,000 by 1949 to 48,000 by 1969...”. Stating the numbers this way is more relevant to our study.

L105 “researchers”, who?

We meant researchers from which we used data from in our analysis. With the new organization of the manuscript, this sentence was changed to “Sedimentation is measured over the last 150 years by establishing geochronologies ...”.

L151 but there is also great land loss in other nearby places in the gulf? In general I would encourage the authors to discuss their results in the context of land loss reported in many coastal places

This part of the paper is explaining why we do not see a significant (P value=0.15) increase in average MAR post 1950 in Sabine Bay. The area of the depocenter is increasing but MAR values are staying about the same. This means that sediment supply to Sabine Bay must be increasing, but we don't observe that because sedimentation is distributed over a larger basin area. Yes, our other sites in Louisiana show an increase in subtidal area, about the same percent increase in Barataria and Terrebonne bays as Sabine Bay, but in LA we observe a doubling in median MAR post 1950. The text was changed and a sentence about Louisiana was added at line 150,

“The average MAR increase in Sabine Bay is not significant (P value=0.15) because of the 24.5% conversion of land between the Sabine and Calcasieu bays to open water during 1956-201036. This increase in area of the depocenter was in proportion to the increase in sediment supplied by the small 30,000 km² coastal-plain watershed. Like Sabine Bay, after 1950, 25.9% and 24.5% of land was converted to open water in Barataria basin and Terrebonne basin, respectively³⁶. At those sites, average MAR more than doubled post 1950 because increased sediment supply to the depocenters exceeded the increase in subtidal areas.”

L200: reference?

We added reference to Figure 6 here.

L212: reference 60 (church and white) does not really validate this statement I believe? Is there a statistical test the authors can refer to which shows that variability in RSLR does not explain the observed MAR/SAR trends

Church and White show an acceleration in SLR of 0.013 +/- 0.006 mm yr⁻² or 0.017 +/- 0.007 mm yr⁻² 1870-2001 CE (quadratic versus comparing linear regressions), but most of this acceleration occurred at the end of the 19th century and early 20th century rather than a smooth acceleration over the whole period. In their abstract it states, “a reconstruction of global sea level using tide-gauge data from 1950 to 2000 indicates a larger rate of rise after 1993 and other periods of rapid sea-level rise but no significant acceleration over this period.” Between 1930 and 1960 SLR is 2.5 mm/yr. It is only after 1993 that rates of SLR increase to 3 mm/yr. We clarified the statement, which now reads, “The acceleration in global SLR ~1880-1930 must have contributed to the increase in accommodation⁷, and global SLR affected each site differently over the last 50 years due to spatial variability in land movement, meteorology, and climate change⁶⁰.”

L231: I am unclear what exactly the supposed difference is between coastal depocenters and deltaic coasts.

The sentence reads, “This is clearly happening along the shorelines of deltaic coasts that are rapidly eroding¹⁹ but may not be the case in subtidal settings of coastal depocenters.” The shorelines of deltaic coasts are not depocenters because they prograde and retrograde in response to a variety of processes (seasonal floods, avulsion events, climate change, etc.). A coastal depocenter is located away from shorelines. We include river-dominated margins in our compilation, but sites are always away from shorelines, hence, components of deltas can be considered coastal depocenters over the 100-year time scale we are addressing in this paper, such as Barataria and Terrebonne bays.

L252: “below”? in general the methods section is also rather unclear and could use better subdivision into separate section.

That reference to information “below” was deleted and we include an additional subsection in the methods section. We also reorganized parts of the methods section.

L171 I would say that an increase in wave energy would lead to enhanced accommodation space. I am probably mistaken because I find the term accommodation space a little confusing, but I would think that increasing waves would lead to more efficient export of sediment from the coastal zone, and hence lead to more space for sediment to be deposited?

An increase in wave or current energy impacting the bottom of a depocenter would increase the depth of erosion. During that period of higher energy there is not more space for sediments to accumulate. During a subsequent period of quiescence, deposition on the bed will be promoted.

Reviewers' comments:

Reviewer #1 (Remarks to the Author):

Rodregueze review 2

Thank you for the opportunity to re-review the manuscript by Rodriguez et al. I recognize and appreciate the efforts the authors have made to revise the initial submission of the manuscript. Their revised manuscript has incorporated many of the edits suggested by me and another anonymous reviewer. In particular, the authors have done an exceptionally good job of addressing the influence of the Sadler effect on their model interpretations. A prime example of this is their explanation of how they used core lithology to examine for substantial hiatuses or changes in sedimentation patterns. The changes have made this manuscript stronger for demonstrating that there is a likely increase in sediment accumulation since the mid 20th century in many of the coastal depocenters that they studied. The other reviewer raised a valid concern in that it would be worthwhile to document the other sources of sediment. My opinion is it is beyond the scope of a short Nature Communications manuscript to explain how & why it happened at each depocenter. Doing so would require a detailed explanation of land use changes in each watershed, how these changes would impact the type/composition of sediment that accumulates within each, and then examining the sediment cores for the corresponding change in composition. The authors' changes address this.

1) Whereas these changes have improved some of the underlying major concerns that I had with the initial manuscript, there are still some inconsistencies within the paper that I feel need to be addressed. To be honest, I am still confused as to the purpose of the paper. Is it to document that sediment accumulation rates have increased in coastal depocenters due to human modification of watersheds since 1950? Or is to test the assumption that dams/impoundments should have reduced sediment delivery, and thus accumulation rates, in coastal watersheds? Testing these two requires different sets of cores & settings. The first requires comparing similar watersheds that have are "pristine" with those that are modified. The second requires examining watersheds that have dams/impoundments and looking for a change in estuarine sedimentation that occurs well after the dams/impoundments have been built.

Given the number of times the word "dam" pops up in this paper, I assume the purpose is the latter: "Line 88: The assumption that coastal depocenters are sediment starved and are experiencing a decrease in sedimentation from the construction of dams upstream needs to be tested". However, on line 106, it states "While many of the smaller watersheds have no dams, the larger watersheds are severely modified by dam construction, which peaked after 1950 and reduced fluvial suspended sediment concentrations away from the coast." If they have no dams, why are they included in a paper to demonstrate that coastal watershed sedimentation has increased even though, hypothetically, dams have trapped sediment?

Some examples of watersheds lacking dams in this paper include: Nastapoka Sound, Hudson Bay, Canada (Jolivel et al., 2015); Jamaica Bay, Queens (Renfro et al., 2016); Chetco River, Oregon (Core CHT; Wheatcroft and Sommerfield, 2005). It appears from the author's rebuttal document and my impressions of the manuscript is that these sites were included because: 1) they are coastal; 2) they have enough data Pb-210 activity data points to show that there has been a human-caused increase in sedimentation rates since 1950 based on the CRS model. They state in their rebuttal comment #5 "Yes, other factors like changes in weather can modify sedimentation, but even Sommerfield et al., 2002 recognized that human impacts contributed to the increase in sedimentation that they documented for the Eel River shelf... and the pericontinental shelf of Hong Kong (Owen and Lee, 2004), where the observed increased input from land-sourced material reflects deforestation and expanding human impacts during periods of major land-use change."

As I see it, a paper that seeks to test whether coastal sedimentation has changed in watersheds that contain dams should limit their results to watersheds that actually have dams within them that could trap sediment. Rather, if the authors' intent in this manuscript is to demonstrate that coastal depocenters have seen an increase in sedimentation since 1950 because of human modifications to watershed erodibility, they should state that in the title, the abstract, and the introduction and let the paper be reviewed with that focus.

I feel the authors have made a strong case that north America estuarine watersheds that contain dams/impoundments do show an increase in sedimentation since 1950. I would recommend that the authors only include watersheds that have had dams or impoundments constructed within the post-1950 time period of interest described by the authors. They should include a column within their Supplementary Data Table that lists when dams/impoundments were established for each watershed.

2) I encourage the authors to discuss their results in light of the recent paper in Nature by Nienhuis et al., "Global-scale human impact on delta morphology has led to net land area gain" doi.org/10.1038/s41586-019-1905-9. The paper by Nienhuis uses a modelling approach to address many of the same questions raised in this manuscript under review, but the paper here under review provides much needed model validation, and therefore is a good complementary work, worthy of publication.

3) The authors state on line 98 that "We targeted shallow (<30 m) subtidal depocenters (generally areas of sediment focusing) as archives of regional changes in coastal sedimentation.". This is not an accurate statement because many of the continental shelf coring sites chosen are in water depths deeper than 30 m (e.g., CHT was in 100 m; Gulf of Tehuantepec was in 67 m; Core S0304SC-5 was in 80 m).

I still consider the use of continental shelf cores less than convincing when testing the influence of dams on changes in sedimentation rates at their location. Mid-to-outer shelf core locations have been shown to decouple from direct fluvial input because of along-shelf sediment dispersal (e.g., Po River-Adriatic, Gulf of Papua, Gulf of Alaska). More importantly here is the use of the CRS model to establish temporally varying sedimentation rates for continental shelf cores. The authors of this manuscript note that these shelf cores were initially modeled with the constant sedimentation/constant initial concentration approach, which is used because this model has boundary conditions that are more suitable for open marine settings. The Constant Rate of Supply (CRS) model assumes: 1) that all the excess Pb-210 activity is scavenged from the atmosphere/water column; 2) that the rate of supply of excess Pb-210 activity does not change over time. These boundary conditions hold for soils, marshes, and freshwater lakes, where the primary excess activity input source is the atmosphere and overall particle concentrations are high enough to scavenge the deposited activity (i.e., original application by Appleby and Oldfield). However, neither of these boundary conditions can be satisfied for the open continental shelf, where upwelling and downwelling of dissolved xsPb-210-rich and xsPb-210-poor water masses is common and violates condition #2. Because the CRS model results in higher apparent sedimentation rates when the excess Pb-210 activity decreases in a core interval, an alternative explanation to the observations in this paper is that higher apparent sedimentation rates since ~1950 have been caused by a reduction in the dissolved excess Pb-210 activity along the dispersal route of the particle from soil to mid-shelf depocenter. This is much less likely to occur in coastal estuaries with reduced ocean water mass input, so this CRS approach is more valid for these settings, in my opinion. Bioturbation can also alter the depth profile of excess activity, which is often 10-20 cm deep in some shelf sites. This can lead to an apparent period of higher or lower sedimentation rates depending on the actual activity that is being mixed. Because of these various possible complications, I continue to urge the authors to consider omitting their continental shelf sites.

4) I agree with the other reviewer that the original introduction was too long. The rewrite is better, but I still feel that the paragraph starting on line 47 "The bathymetry of coastal depocenters is an important determinant..." should be moved or omitted. This is all general background information on how sedimentation in coastal depocenters works and should be background information. The introduction flows much better if this entire paragraph is removed from this section. Also, the multiple uses of the vague term "modulated" in this paragraph is distracting to this reader at least. This term should be made more specific if the authors wish to retain it in the final version.

5) The revised manuscript has errors with the citation numbers. It appears from the track changes document that many citations were removed from the primary manuscript (probably to make room for the other text). Their original call-out numbers are still used in the text and tables and should be updated (e.g., #27 in original manuscript; Renfro et al.). Also, some of the citations are incomplete (e.g., Alexander and Lee is missing the title of their paper). Double check all references, please.

Sincerely,
John M. Jaeger

Reviewer #2 (Remarks to the Author):

I have reviewed the revised manuscript "coastal sedimentation doubled during the Anthropocene despite river dams", where Rodriguez et al compile sediment core accumulation rate studies to describe recent changes to sediment rates across the US and Canada.

The study has improved considerably. I request mostly minor revisions for clarity and completeness.

However, I have one major remark that I think should be addressed. At several instances in the text, it is alluded to that the origin of the increase in MAR and SAR could result from (1) eroding shorelines, (2) intertidal habitat, and (3) relative sea level rise (presumably moving sediment onshore; it would not have an effect on sediment delivery to the coast). Any of these sources of sediment would invalidate the conclusions of this study. I therefore think that the authors should address (statistically) that in most of the sites these sources are not dominant. If they are dominant, then a vertical core in the (for example) eroding intertidal habitat would (presumably) show a decrease in MAR over the same period. The same holds for places where the authors suggest an increase in relative sea level had resulted in the increase in MAR/SAR. In those places, the sediment could have easily originated from another location further offshore, thus invalidating the conclusion of this study.

Minor comments

L25 assuming no feedbacks

L40 than -> compared to

L45 I would say more than some. A lot of the relative sea level rise analyses of the Louisiana coast are derived from sedimentation (Jankowski et al., Nature Communications, for example).

L66 perhaps once something is subaerial, but over the past ~6000 years I would say coastal areas generally have gained land and are therefore not in equilibrium w/ RSLR.

L92 perhaps more explicitly state this hypothesis as the main aim of the study.

L141 so the site in florida had a disproportionate increase from 1950-1980?

L138 vs. L145 "every" vs "most". I don't understand. Every site shows a increase through time, but sabine bay does not? I assume the difference is that one site the increase is not statistically significant. I would however then not call that a "general increase" (L139) either.

L150 I don't understand. The increase here is not significant because of the land loss? Then thereafter the authors seem to suggest a link; such that the land loss was responsible ("in

proportion") for the increase in sediment deposition (L152)? Independent sediment flux data that is supplied to the bay would be needed to substantiate this claim.

L159 I would reverse these two sentences to make the story easier to read and understand. First state the observations and then their implications. (e.g. we do not see variation in those factors, implying that MAR increase is caused by an increase in the sediment flux).

L168 replace "recent" with "additional"

L179 how should I envision a sediment flux in m/yr?

L221 SLR alone would increase accommodation, correct? You don't necessarily need acceleration of SLR.

L250 what exactly is the difference between a coastal depocenter and a deltaic coast? I realize I asked this in my earlier review. The authors responded to my question, but did not address it in their paper. In general, I would like to see a clear definition of a coastal depocenter. Shorelines of deltaic coasts (on appropriate timescales) must be depositing as well for low RSLR, just like other components of these coasts.

L265 for clarity and international audience, rewrite to Louisiana and Texas coastal areas.

We have addressed the criticisms provided in this second round of reviews by the Editor, Dr. Jaeger, and Reviewer 2. All comments resulted in us making a change to the manuscript text or supplementary data and as a result the manuscript is significantly improved. The reviewers must have invested a significant amount of time preparing their thoughtful criticisms and we thank them for sharing ideas with us.

Reviewer #1 Dr. John M. Jaeger:

Thank you for the opportunity to re-review the manuscript by Rodriguez et al. I recognize and appreciate the efforts the authors have made to revise the initial submission of the manuscript. Their revised manuscript has incorporated many of the edits suggested by me and another anonymous reviewer. In particular, the authors have done an exceptionally good job of addressing the influence of the Sadler effect on their model interpretations. A prime example of this is their explanation of how they used core lithology to examine for substantial hiatuses or changes in sedimentation patterns. The changes have made this manuscript stronger for demonstrating that there is a likely increase in sediment accumulation since the mid 20th century in many of the coastal depocenters that they studied. The other reviewer raised a valid concern in that it would be worthwhile to document the other sources of sediment. My opinion is it is beyond the scope of a short Nature Communications manuscript to explain how & why it happened at each depocenter. Doing so would require a detailed explanation of land use changes in each

watershed, how these changes would impact the type/composition of sediment that accumulates within each, and then examining the sediment cores for the corresponding change in composition. The authors' changes address this.

1A) Whereas these changes have improved some of the underlying major concerns that I had with the initial manuscript, there are still some inconsistencies within the paper that I feel need to be addressed. To be honest, I am still confused as to the purpose of the paper. Is it to document that sediment accumulation rates have increased in coastal depocenters due to human modification of watersheds since 1950? Or is to test the assumption that dams/impoundments should have reduced sediment delivery, and thus accumulation rates, in coastal watersheds? Testing these two requires different sets of cores & settings. The first requires comparing similar watersheds that have are “pristine” with those that are modified. The second requires examining watersheds that have dams/impoundments and looking for a change in estuarine sedimentation that occurs well after the dams/impoundments have been built.

We do want the purpose of the study to be presented clearly. With the changes to the Introduction we made, as indicated in Dr. Jaeger's Comment 4 (below), the purpose statement now comes much earlier. We rewrote the paragraph that presents the aim of this study, which begins on Line 66, and is reproduced here:

“The objective of this study is to test the assumption that North American coastal depocenters are sediment starved from the damming of rivers. For this study, we define coastal depocenters as subtidal basins away from shorelines that are net depositional. We hypothesize that reduced suspended-sediment delivery to the coast from impoundments will be recorded in coastal depocenters as decreasing sedimentation rates over the last 100 years. Alternatively, if sediment sources positioned downstream of dams offset sediment lost to reservoirs, then that will be recorded as constant or increasing sedimentation rates. While dams certainly interrupt the river-sediment transport pathway, that does not necessarily indicate that sedimentation in coastal depocenters is less than it was, and that water depths are increasing with SLR.”

The largest human modifications made to watersheds are the construction of impoundments and changes in land cover. Impoundments have undeniably trapped sediment, as shown by many previous studies that look at river gauges. We argue that there are additional sources downstream that have overcompensated for those losses because we see a doubling in sedimentation at most of the 25 sites.

1B) Given the number of times the word “dam” pops up in this paper, I assume the purpose is the latter: “Line 88: The assumption that coastal depocenters are sediment starved and are experiencing a decrease in sedimentation from the construction of dams upstream needs to be tested”.

We have changed this statement for clarification as to what the objective of our study is. That text was removed and replaced with the text pasted in the previous response.

1C) However, on line 106, it states “While many of the smaller watersheds have no dams, the larger watersheds are severely modified by dam construction, which peaked after 1950 and reduced fluvial suspended sediment concentrations away from the coast.” If they have no dams, why are they included in a paper to demonstrate that coastal watershed sedimentation has increased even though, hypothetically, dams have trapped sediment?

The sentence on Line 106 was supposed to be read in context with the previous sentence. The entire phrase was written as follows:

“Each depocenter receives sediment from multiple watersheds, including both small (< 250 km²) watersheds that are isolated to coastal regions and larger watersheds that extend into piedmont or high-relief areas. While many of the smaller watersheds have no dams, the larger watersheds are severely modified by dam construction, which peaked after 1950 and reduced fluvial suspended sediment concentrations away from the coast.”

Our intention was to communicate that each depocenter receives sediment from multiple watersheds and some of the watersheds that deliver sediment to a depocenter do not have dams, but others have dams. We revised the text to make it clearer. It now reads on Line 109, “The watersheds that contribute suspended sediment to a depocenter are modified by humans to different degrees with some of the smaller watersheds having few or no dams and the larger watersheds being severely modified by impoundments.”

1D) Some examples of watersheds lacking dams in this paper include: Nastapoka Sound, Hudson Bay, Canada (Jolivel et al., 2015); Jamaica Bay, Queens (Renfro et al., 2016); Chetco River, Oregon (Core CHT; Wheatcroft and Sommerfield, 2005). It appears from the author’s rebuttal document and my impressions of the manuscript is that these sites were included because: 1) they are coastal; 2) they have enough data Pb-210 activity data points to show that there has been a human-caused increase in sedimentation rates since 1950 based on the CRS model. They state in their rebuttal comment #5 “Yes, other factors like changes in weather can modify sedimentation, but even Sommerfield et al., 2002 recognized that human impacts contributed to the increase in sedimentation that they documented for the Eel River shelf... and the pericontinental shelf of Hong Kong (Owen and Lee, 2004), where the observed increased input from land-sourced material reflects deforestation and expanding human impacts during periods of major land-use change.”

Apparently, we misinterpreted Dr. Jaeger’s main point of comment #5 in the previous review. We thought Dr. Jaeger was using the Eel River as an example of a site where changes in sediment delivery to a continental shelf site had nothing to do with anthropogenic alteration of sediment yields and therefore, we should avoid the continental shelf setting. We responded that there were changes in delivery from human impacts to the Eel River and that many other studies of sedimentation in shelf settings noticed changes. Now, we understand that Dr. Jaeger was specifically referring to dams as the anthropogenic alteration.

1E) As I see it, a paper that seeks to test whether coastal sedimentation has changed in watersheds that contain dams should limit their results to watersheds that actually have

dams within them that could trap sediment. Rather, if the authors' intent in this manuscript is to demonstrate that coastal depocenters have seen an increase in sedimentation since 1950 because of human modifications to watershed erodibility, they should state that in the title, the abstract, and the introduction and let the paper be reviewed with that focus.

We now define depocenters more clearly on Line 67, "For this study, we define coastal depocenters as subtidal basins away from shorelines that are net depositional." Depocenters are not uniquely tied to one watershed and on Line 102 we include "The depocenters primarily receive suspended sediment from numerous watersheds, including both small (< 250 km²) watersheds that are isolated to coastal regions and larger watersheds that extend into piedmont or high-relief areas." Estuaries and continental shelves have multiple rivers that discharge into them. For example, Mobile Bay is primarily fed by the Mobile and Tombigbee Rivers (lots of dams), but there is also the smaller Dog, Fowl, Bon Secour, Fish, and Magnolia rivers that contribute sediment to the depocenter and some of those rivers are lacking dams. We are not just cherry-picking Mobile Bay with this example, multiple rivers of various sizes with different levels of human modifications supply sediment to all the depocenters. The proportion of sediment being deposited in the depocenters from each possible source is unknown and beyond the scope of this study, as Dr. Jaeger also recognizes in his initial review comment, above.

Our sites have different levels of human modification to sediment source areas and it is true that some sites are more directly influenced by watersheds with dams than other sites. Dr Jaeger correctly recognizes Nastapoka Sound and the Pacific Northwest margin as two depocenters included in our study that are likely the furthest removed from watersheds that contain dams; however, those two sites and Jamaica Bay are not entirely isolated from dams (see below). Considering each site shows basically the same trend, and we never combine sites into one regression or mean, we can delete sites and it will have no impact on our findings, except in this case reducing our sample size from 25-22. Our only reluctance is that there needs to be solid justification for deleting sites and we worry that deleting sites will remove diversity from the dataset. Jamaica Bay is our most urban example, Nastapoka Sound is the only example we have where degradation of permafrost from climate change is impacting sedimentation (now called out more explicitly on Lines 192-195, "Rapid permafrost decay in river catchments from warming climate is an additional source of sediment to high latitude depocenters and contributed to the MAR increase in Nastapoka Sound") and the Pacific Northwest Margin site represents one of only 3 West Coast sites. We added text to the Methods section Lines 307-309 stating, "Additionally, sites were selected to maximize the diversity of North American geologic and climatic settings included in the study." We agree that it is important to address that there is variability in connectivity between dammed rivers and depocenters among sites and to address that, we added more information about sediment sources to the sites. On Lines 113-117 we added,

"The connectivity of watersheds that contain dams to a depocenter varies among the sites. For example, the Nastapoka Sound depocenter is positioned 350 km down drift from the nearest outlet of a dammed river, in contrast to the 13

depocenters located in drowned river mouth estuaries that are more confined repositories for suspended sediment from dammed rivers.”

We appreciate the time Dr. Jaeger spent examining our sites in detail. Below are summaries of our understanding of how Jamaica Bay, the Pacific Northwest Margin, and Nastapoka Sound depocenters are influenced by dams.

Jamaica Bay- The depocenter receives tidal water from the adjacent and relatively large Hudson River, which has many dams. The Jamaica Bay watershed is an urban landscape and has a long history of damming, but at present most of the area is completely impervious. The locations of old reservoirs are now parks, making it difficult to recognize where dams were located from imagery alone. The National Inventory of Dams is a useful resource for identifying operational dams in the United States (<https://nid.sec.usace.army.mil/>). The Ridgewood Reservoir Dam is in the Jamaica Bay watershed (National Inventory of Dams ID NY00160), was created in 1891 to supply water to New York City, and was last inspected on October 22, 2015. Springfield Park was created around an old impounded Mill Pond, also located within the Jamaica Bay watershed. Baisley Pond is another reservoir in the watershed that was constructed when three streams were dammed and was continually expanded as New York City grew around it and the reservoir became an important water resource for the growing population. It is currently Baisley Pond Park. The drainage basin areas around Jamaica Bay, including the channels, are highly modified by urban development.

Pacific Northwest Margin- Core CHT was collected on the middle continental shelf offshore of the Chetco River. In addition to the Chetco River watershed containing a dam, it is unclear what the contribution of sediment to the site is from currents that flow along the coast and some rivers north and south of the Chetco are also dammed. Ferry Creek is a tributary to the Chetco River (it is part of the watershed) and the Ferry Creek dam (National Inventory of Dams ID OR00437), located 1 km from the confluence with the Chetco River, was completed in 1966.

Nastapoka Sound- Hudson Bay has a general counter-clockwise circulation and the eastern margin around Nastapoka Sound is characterized by a strong coastal current that flows from south to north (Saucier et al. 2004; St-Laurent et al., 2011). The La Grande drainage basin to the south discharges into James Bay and is highly regulated with numerous dams and hydroelectric facilities. Some portion of the suspended sediment load to James Bay is transported northward into Hudson Bay (d'Anglejan, 1980; Kuzyk et al., 2009). Jolivel et al. (2015) also recognized that southern source of inputs to the depocenter,

“Nastapoka Islands likely act as a barrier by holding freshwater and nutrient inputs released by numerous rivers flowing into James Bay and eastern Hudson Bay (Hudon et al. 1996; Déry et al. 2005). Moreover, the counter-clockwise water circulation and strong coastal currents may limit offshore dispersion of river inputs.”

Saucier, F., Senneville, S., Prinsenber, S., Roy, F., Smith, G., Gachon, P., Caya, D., Laprise, R., 2004. Modelling the sea ice-ocean seasonal cycle in Hudson Bay, Foxe Basin and Hudson Strait, Canada. *Climate Dynamics* 23, 303-326.

St-Laurent, P., Straneo, F., Dumais, J.-F., Barber, D.G., 2011. What is the fate of the river waters of Hudson Bay? *Journal of Marine Systems* 88, 352-361.

Biksham, G., d'Anglejan, B., 1989. Rate of sedimentation and geochemistry of southeastern Hudson Bay, Canada. *Sediment and the environment: International Association of Hydrological Sciences Publication* 184, 27-36.

Kuzyk, Z.Z.A., Macdonald, R.W., Johannessen, S.C., Gobeil, C., Stern, G.A., 2009. Towards a sediment and organic carbon budget for Hudson Bay. *Marine Geology* 264, 190-208.

Hudon, C., Morin, R., Bunch, J., Harland, R., 1996. Carbon and nutrient output from the Great Whale River (Hudson Bay) and a comparison with other rivers around Quebec. *Canadian Journal of Fisheries and Aquatic Sciences* 53, 1513-1525.

Déry, S.J., Stieglitz, M., McKenna, E.C., Wood, E.F., 2005. Characteristics and Trends of River Discharge into Hudson, James, and Ungava Bays, 1964–2000. *Journal of Climate* 18, 2540-2557.

1F. I feel the authors have made a strong case that north America estuarine watersheds that contain dams/impoundments do show an increase in sedimentation since 1950. I would recommend that the authors only include watersheds that have had dams or impoundments constructed within the post-1950 time period of interest described by the authors. They should include a column within their Supplementary Data Table that lists when dams/impoundments were established for each watershed.

The depocenters are not receiving sediment from only one watershed. This is a better assumption for deltas, but not for the sites included in this study. It is impossible to only include watersheds that have had dams or impoundments constructed within the post-1950 time period in the study because each depocenter is receiving sediment from watersheds that are dammed and watersheds that don't have dams, and rivers are modified by many dams constructed throughout the last century both pre- and post-1950. The main point of this paper is that sedimentation in coastal depocenters is keeping up with sea-level rise and has shown a doubling despite all the sites being influenced by dams (some more than others). These results run counter to the notion that dams are starving the coast of sediment. Including a table of dam construction in all the possible watersheds that could be delivering sediment to the depocenters might be misleading considering we don't know the relative contribution of sediment each river provides to the depocenter. We added text on Lines 107-109 presenting that information, "A sediment budget for each depocenter that quantifies the relative contribution of sediment sources to the sedimentary record neither exists for the sites nor can be constructed from the existing data sets."

2) I encourage the authors to discuss their results in light of the recent paper in Nature by Nienhuis et al., "Global-scale human impact on delta morphology has led to net land area gain" doi.org/10.1038/s41586-019-1905-9. The paper by Nienhuis uses a modelling approach to address many of the same questions raised in this manuscript under review,

but the paper here under review provides much needed model validation, and therefore is a good complementary work, worthy of publication.

The Nienhuis et al., (2020) paper presents a very interesting modeling study looking at how the morphology of 11,000 deltas (defined broadly) worldwide has been affected by river damming and deforestation. Their modeling results show a net gain in land area over the past 30 years and attribute that to human modification of watersheds (deforestation), noting that new source of sediment overwhelmed losses due to impoundments. We now refer to that new study on lines 261-263, “This is clearly happening along the shorelines of deltaic coasts that are rapidly eroding but may not be the case in subtidal settings of coastal depocenters or ~ 90% of deltas, worldwide (Nienhuis et al., 2020).”

3A) The authors state on line 98 that “We targeted shallow (<30 m) subtidal depocenters (generally areas of sediment focusing) as archives of regional changes in coastal sedimentation.” This is not an accurate statement because many of the continental shelf coring sites chosen are in water depths deeper than 30 m (e.g., CHT was in 100 m; Gulf of Tehuantepec was in 67 m; Core S0304SC-5 was in 80 m).

This was supposed to be a general statement about the depocenters, not the specific core locations. We deleted that wording from the text and it now reads on lines 83-86, “We used depocenters as archives of regional changes in coastal sedimentation, targeting sites where variations in those depositional, erosional, and mixing processes that form sedimentary records were generally constant through time (supplemental information).” To further address Dr. Jaeger’s comment, we created a new column in supplemental data that presents water depth at each core location and refer to that in the text on Lines 80-83, “The compilation of sites spans a wide range of geologic and climatic settings and includes estuaries (18; average core water depth=10 m) and inner continental shelves (7; average core water depth=49 m; Fig. 1; supplementary Table 1).”

3B. I still consider the use of continental shelf cores less than convincing when testing the influence of dams on changes in sedimentation rates at their location. Mid-to-outer shelf core locations have been shown to decouple from direct fluvial input because of along-shelf sediment dispersal (e.g., Po River-Adriatic, Gulf of Papua, Gulf of Alaska). More importantly here is the use of the CRS model to establish temporally varying sedimentation rates for continental shelf cores. The authors of this manuscript note that these shelf cores were initially modeled with the constant sedimentation/constant initial concentration approach, which is used because this model has boundary conditions that are more suitable for open marine settings. The Constant Rate of Supply (CRS) model assumes: 1) that all the excess Pb-210 activity is scavenged from the atmosphere/water column; 2) that the rate of supply of excess Pb-210 activity does not change over time. These boundary conditions hold for soils, marshes, and freshwater lakes, where the primary excess activity input source is the atmosphere and overall particle concentrations are high enough to scavenge the deposited activity (i.e., original application by Appleby and Oldfield). However, neither of these boundary conditions can be satisfied for the open continental shelf, where upwelling and downwelling of dissolved xsPb-210-rich and xsPb-210-poor water masses is

common and violates condition #2. Because the CRS model results in higher apparent sedimentation rates when the excess Pb-210 activity decreases in a core interval, an alternative explanation to the observations in this paper is that higher apparent sedimentation rates since ~1950 have been caused by a reduction in the dissolved excess Pb-210 activity along the dispersal route of the particle from soil to mid-shelf depositor. This is much less likely to occur in coastal estuaries with reduced ocean water mass input, so this CRS approach is more valid for these settings, in my opinion. Bioturbation can also alter the depth profile of excess activity, which is often 10-20 cm deep in some shelf sites. This can lead to an apparent period of higher or lower sedimentation rates depending on the actual activity that is being mixed. Because of these various possible complications, I continue to urge the authors to consider omitting their continental shelf sites.

Continental shelf sites are less confined settings than estuaries and we agree that it is important to address that in the paper. We now state that continental shelf sites receive some sediment from the along-shelf sediment dispersal system and are less connected to dammed rivers than estuaries. Lines 104-107 were added; “The sites on the middle continental shelf (sites 23 and 25) are more marine influenced than the other sites and receive some sediment from the along-shelf sediment dispersal system, making their sediments an indirect recorder of changes to watersheds.” No coastal depositor receives sediment from only one source.

Two sites in this study (Santa Clara Shelf and Pacific Northwest Margin) are located on the middle continental shelf. These locations add a possible complication to modeling Pb-210 profiles to determine sedimentation rates because their outer shelf location presents the potential for an additional source of excess Pb-210 resulting from upwelled waters. It is possible that upwelled waters could bring a variable Pb-210 source (higher or lower) than supplied by atmospheric deposition and therefore violate the assumptions of constant Pb-210 activity of sediment supplied to the bottom (CIC model used in the original publications) or of a constant flux of Pb-210 to the site (CRS model used here). How upwelling might compromise geochronology models would depend on (a) the concentration of Pb-210 in the upwelled waters; and (b) the frequency of upwelling.

No information exists regarding Pb-210 concentrations in upwelled waters at the two sites. At both sites, upwelling is seasonal occurring during the summer months (Hickey, 1998; Hickey et al., 2016). SAR values reported in the original publications using the CIC model are 2.6 and 5.8 mm yr⁻¹, respectively for the Santa Clara Shelf and the Pacific Northwest Margin sites. Our average CRS-derived SAR values are lower (0.8 and 2.1 mm yr⁻¹, respectively). Cores were subsampled at 1-cm intervals at both sites. Therefore, based on the CIC model, each interval sampled integrated over approximately 2 to 4 upwelling cycles. The frequency of variability for Pb-210 supply due to upwelling to each site would be less than the minimum time interval represented by each subsample and should therefore have minimal impact on the centimeter by centimeter geochronology established.

We agree that it is important to address possible effects from upwelling in the text, but the presence of upwelling and the contribution of sediment from along-shore sources should not exclude those sites from our study. Including mid-shelf sites adds to the growing body of

evidence disproving sediment starvation, but we should present information in the text outlining that those sites are a bit different from the other depocenters. On Lines 127-129 we added the text, “A possible complication for establishing ^{210}Pb geochronologies may arise at sites along the Pacific margin (sites 23 and 25) where the source of ^{210}Pb from upwelling waters may vary (see methods).” In the methods section we added a paragraph (like what is written above) to explain more about the influence of upwelling on the geochronologies (Lines 348-361).

Hickey, B.M. (1998). Coastal oceanography of western North America from the tip of Baja California to Vancouver Island. In: Robinson, A.R., Brink, K.H. (Eds.), *The Sea*, vol. 11. John Wiley & Sons, Inc, New York, pp. 345–393.

Hickey, B., S. Geier, N. Kachel, S. Ramp, P. M. Kosro, and T. Connolly (2016), Along coast structure and interannual variability of seasonal midshelf water properties and velocity in the Northern California Current System, *J. Geophys. Res. Oceans*, 121, 7408–7430,

4) I agree with the other reviewer that the original introduction was too long. The rewrite is better, but I still feel that the paragraph starting on line 47 “The bathymetry of coastal depocenters is an important determinant...” should be moved or omitted. This is all general background information on how sedimentation in coastal depocenters works and should be background information. The introduction flows much better if this entire paragraph is removed from this section. Also, the multiple uses of the vague term “modulated” in this paragraph is distracting to this reader at least. This term should be made more specific if the authors wish to retain it in the final version.

We agree that the introduction reads much better without that paragraph. We moved a shortened version of that text to the second paragraph of the results and discussion section (Lines 89-101) and deleted the word “modulated”.

5) The revised manuscript has errors with the citation numbers. It appears from the track changes document that many citations were removed from the primary manuscript (probably to make room for the other text). Their original call-out numbers are still used in the text and tables and should be updated (e.g., #27 in original manuscript; Renfro et al.). Also, some of the citations are incomplete (e.g., Alexander and Lee is missing the title of their paper). Double check all references, please.

Yes, regrettably the references in Supplementary Table 1 reflected version 1 of the manuscript and were updated. It is now a stand-alone document with references listed below the table. The title was added to the Alexander and Lee reference in supplementary information. We double checked all references.

Reviewer #2 (Remarks to the Author):

I have reviewed the revised manuscript “coastal sedimentation doubled during the Anthropocene despite river dams”, where Rodriguez et al compile sediment core

accumulation rate studies to describe recent changes to sediment rates across the US and Canada.

The study has improved considerably. I request mostly minor revisions for clarity and completeness.

1. However, I have one major remark that I think should be addressed. At several instances in the text, it is alluded to that the origin of the increase in MAR and SAR could result from (1) eroding shorelines, (2) intertidal habitat, and (3) relative sea level rise (presumably moving sediment onshore; it would not have an effect on sediment delivery to the coast). Any of these sources of sediment would invalidate the conclusions of this study. I therefore think that the authors should address (statistically) that in most of the sites these sources are not dominant. If they are dominant, then a vertical core in the (for example) eroding intertidal habitat would (presumably) show a decrease in MAR over the same period. The same holds for places where the authors suggest an increase in relative sea level had resulted in the increase in MAR/SAR. In those places, the sediment could have easily originated from another location further offshore, thus invalidating the conclusion of this study.

There are multiple sources of sediment to the depocenters. We were not clear that eroding shorelines and intertidal habitat represent a source of sediment to the depocenters; however, those sources cannot explain the magnitude of the increase we observe at the sites, with the exception being coastal Louisiana. The problem is scale. If we continue with Mobile Bay as an example (from responses above) and apply a simple box model, there is ~140,000 m of shoreline around the depocenter and if we assume all of that shoreline is eroding at an average rate of 0.5 m yr⁻¹ and to an average depth of 1.0 m, that would produce ~70,000 m³ of sediment every year, but the area of the depocenter is 970 million m². Even if you assume all that material is silt and clay, 100% retention in the depocenter and persistent erosion for 50 years, then sediment from shoreline erosion would only fill 2% of the accommodation from SLR alone, distributed equally on the bay floor. We crafted new text on lines 185-192 to clarify.

“The sediments from eroding shorelines and degrading intertidal habitats are additional secondary sources to coastal depocenters, but for most of the sites this source is too low to explain the increase in MAR given the disproportionately large area of the depocenter relative to the area of shoreline loss (orders of magnitude difference) and that shoreline erosion increases the size of the depocenter. The exception being the coastal depocenters of Louisiana where these sources promoted the MAR increase post 1950 despite the distinct reduction in Mississippi River flow and suspended-sediment flux to the margin, which contributed to the loss of wetlands.”

Sea-level rise impacts the sites by creating accommodation, not by moving sediment onshore. At most of the sites SLR and SAR match, supporting previous work by Nichols (1989). To address identifying the relative contribution of each sediment source to the site, we write on Lines 107-109 “A sediment budget for each depocenter that quantifies the relative contribution of sediment

sources to the sedimentary record neither exists for the sites nor can be constructed from the existing data sets.” Further along in the paper we state on lines 195-198, “This study falls short of identifying the relative additional contributions of specific downstream sediment sources to coastal depocenters; a necessary next step...” Fingerprinting the various sediment sources and recognizing their contribution to the sedimentary records is just beyond the scope of this study. That notion is echoed by comments made by Reviewer 1, in Dr. Jaegers initial comment, pasted below.

“The other reviewer raised a valid concern in that it would be worthwhile to document the other sources of sediment. My opinion it is beyond the scope of a short Nature Communications manuscript to explain how & why it happened at each depocenter. Doing so would require a detailed explanation of land use changes in each watershed, how these changes would impact the type/composition of sediment that accumulates within each, and then examining the sediment cores for the corresponding change in composition. The authors’ changes address this.”

Minor comments

L25 assuming no feedbacks

The addition was made.

L40 than -> compared to

The change was made.

L45 I would say more than some. A lot of the relative sea level rise analyses of the Louisiana coast are derived from sedimentation (Jankowski et al., Nature Communications, for example).

We agree and changed the word “some” to “many.”

L66 perhaps once something is subaerial, but over the past ~6000 years I would say coastal areas generally have gained land and are therefore not in equilibrium w/ RSLR.

I think this comment goes back to us not clearly defining what a coastal depocenter is because that subtidal basin is what we are referring to as accreting at the rate of RSLR. This paragraph was moved and now comes after where we define a coastal depocenter on Line 67. Some coastal areas, such as deltas, have gained land over the last 6000 years but subtidal depocenters, like the middle part of estuaries and inner continental shelves have not. Hopefully, by placing our definition of a depocenter earlier in the manuscript addresses this comment.

L92 perhaps more explicitly state this hypothesis as the main aim of the study.

This is a similar comment to one that Dr. Jaeger raised. We have rewritten the paragraph and now explicitly state the objective of our study on Lines 66-75.

L141 so the site in florida had a disproportionate increase from 1950-1980?

We changed the text to "...the most extreme being Florida Bay where sediment MAR increased disproportionately from 1950-1980 with the subsequent decrease due to changes in Everglades water management that increased freshwater delivery..."

L138 vs. L145 "every" vs "most". I don't understand. Every site shows a increase through time, but sabine bay does not? I assume the difference is that one site the increase is not statistically significant. I would however then not call that a "general increase" (L139) either.

Yes, that is correct, the increase is not above the significance threshold for Sabine Bay. We agree, not very clear language here. We changed the text beginning on Line 158 to, "Similarly, average sediment MAR since 1950 increased at most of the sites as compared with average pre-1950 values (median P value= 3.92×10^{-5}), the only exception being Sabine Bay where the increase was not significant (P value=0.15; Fig. 5a; supplementary table 1)." We changed the text (old L139) to "...most sites show a general increase..."

L150 I don't understand. The increase here is not significant because of the land loss? Then thereafter the authors seem to suggest a link; such that the land loss was responsible ("in proportion") for the increase in sediment deposition (L152)? Independent sediment flux data that is supplied to the bay would be needed to substantiate this claim.

We were trying to communicate that for MAR to remain constant with an increase in area of the depocenter, that would require an increase in sediment flux, but it is written too definitively as reviewer 2 points out and we don't have independent sediment-flux data to justify using "in proportion." On Lines 160-163, the text now reads, "The land between Sabine and Calcasieu bays is being converted to open water, 24.5% submerged during the period 1956-2010. For the MAR in Sabine Bay to remain constant over this period of depocenter expansion requires an increase in sediment flux."

L159 I would reverse these two sentences to make the story easier to read and understand. First state the observations and then their implications. (e.g. we do not see variation in those factors, implying that MAR increase is caused by an increase in the sediment flux).

This is better. The change was made.

L168 replace "recent" with "additional"

The change was made.

L179 how should I envision a sediment flux in m/yr?

That unit was not intended to represent a flux. The text was changed to "...net transport velocities of < 10 m/yr."

L221 SLR alone would increase accommodation, correct? You don't necessarily need acceleration of SLR.

Yes, SLR alone creates accommodation. We changed the text to “Global SLR must have contributed to the increase in accommodation...”

L250 what exactly is the difference between a coastal depocenter and a deltaic coast? I realize I asked this in my earlier review. The authors responded to my question, but did not address it in their paper. In general, I would like to see a clear definition of a coastal depocenter. Shorelines of deltaic coasts (on appropriate timescales) must be depositing as well for low RSLR, just like other components of these coasts.

The term coastal depocenter is very general and we agree it needs to be defined early in the paper. At the end of the introduction on Line 67 we inserted the sentence, “For this study, we define coastal depocenters as subtidal basins away from shorelines that are net depositional.”

L265 for clarity and international audience, rewrite to Louisiana and Texas coastal areas.

The change was made.

REVIEWERS' COMMENTS:

Reviewer #2 (Remarks to the Author):

Dear Editor,

I have reviewed the revised manuscript and associated response document. The authors have done a good job addressing my concerns; I no longer have any comments.